# *ANGPTL7*, a therapeutic target for increased intraocular pressure and glaucoma

Kavita Praveen[1], Gaurang C. Patel [2], Lauren Gurski[1], Ariane H. Ayer[1], Trikaladarshi Persaud[1], Matthew D. Still [2], Lawrence Miloscio[1], Tavé Van Zyl[2], Silvio Alessandro Di Gioia[1], Ben Brumpton [3,4,5], Kristi Krebs [6], Bjørn Olav Åsvold [3,4,7], Esteban Chen[1], Venkata R. M. Chavali[8], Wen Fury [2], Harini V. Gudiseva[8], Sarah Hyde[9], Eric Jorgenson [1], Stephanie Lefebvre[9], Dadong Li [1], Alexander Li[1], James McIninch[9], Brijeshkumar Patel[2], Jeremy S. Rabinowitz [2], Rebecca Salowe[8], Claudia Schurmann[10], Anne-Sofie Seidelin[11], Eli Stahl [1], Dylan Sun[1], Tanya M. Teslovich[1], Anne Tybjærg-Hansen[11], Cristen Willer [12,13,14], Scott Waldron[9], Sabrina Walley[2], Hua Yang[2], Sarthak Zaveri[2], Regeneron Genetics Center*, GHS-RGC DiscovEHR Collaboration*, Estonian Biobank Research Team*, Ying Hu[2], Kristian Hveem[3,4], Olle Melander[15,16], Lili Milani [6], Stefan Stender[11], Joan M. O'Brien[8], Marcus B. Jones[1], Gonçalo R. Abecasis [1], Michael N. Cantor[1], Jonathan Weyne[2], Katia Karalis[1], Aris Economides [1,2], Giusy Della Gatta[1], Manuel A. Ferreira [1], George D. Yancopoulos[2], Aris Baras [1✉], Carmelo Romano[2✉] & Giovanni Coppola [1✉]

Glaucoma is a leading cause of blindness. Current glaucoma medications work by lowering intraocular pressure (IOP), a risk factor for glaucoma, but most treatments do not directly target the pathological changes leading to increased IOP, which can manifest as medication resistance as disease progresses. To identify physiological modulators of IOP, we performed genome- and exome-wide association analysis in >129,000 individuals with IOP measurements and extended these findings to an analysis of glaucoma risk. We report the identification and functional characterization of rare coding variants (including loss-of-function variants) in ANGPTL7 associated with reduction in IOP and glaucoma protection. We validated the human genetics findings in mice by establishing that *Angptl7* knockout mice have lower (~2 mmHg) basal IOP compared to wild-type, with a trend towards lower IOP also in heterozygotes. Conversely, increasing murine Angptl7 levels via injection into mouse eyes increases the IOP. We also show that acute *Angptl7* silencing in adult mice lowers the IOP (~2–4 mmHg), reproducing the observations in knockout mice. Collectively, our data suggest that ANGPTL7 is important for IOP homeostasis and is amenable to therapeutic modulation to help maintain a healthy IOP that can prevent onset or slow the progression of glaucoma.

A full list of author affiliations appears at the end of the paper.

Glaucoma is a leading cause of irreversible blindness, with a global prevalence of 3.54% in individuals 40–80 years of age, and is projected to affect more than 111.8 million people by 2040[1]. Classified as a neurodegenerative disease, glaucoma is characterized by the progressive loss of retinal ganglion cells in the eye and thinning of the neuroretinal rim of the optic nerve head. Affected individuals present with visual field loss that is accompanied by increased intraocular pressure (IOP) in the majority of cases[2]. Primary open angle glaucoma (POAG) is the most common glaucoma subtype and has highest prevalence in individuals of African ancestry (4.2% prevalence in Africa[1]).

Individuals at greatest risk for POAG are >60 years of age, have a family history of glaucoma, or have high myopia[3–6]. Measurable ocular anatomical and physiological features, including low central corneal thickness (CCT), increased cup-to-disc ratio and high intraocular pressure (IOP)[7] correlate with increased risk for glaucoma and, like glaucoma[8], are highly heritable[4,9–13]. Thus, these quantitative risk factors, when measured on large numbers of individuals, can provide a well-powered dataset for genetic studies to elucidate the etiology of glaucoma risk and progression. The latest genome-wide association study (GWAS) of IOP included more than 130,000 individuals and increased the tally of IOP-associated loci to over 100[14,15]. An earlier GWAS[16] reported that ~89% of loci associated with IOP at a genome-wide significant level showed directionally consistent effects on glaucoma risk, thus reinforcing the utility of quantitative risk factors in gene discovery for glaucoma.

Lowering IOP is the mainstay of all glaucoma therapeutics as IOP continues to be the only modifiable risk factor for the onset or progression of glaucoma. While many effective topical IOP-lowering agents across multiple drug classes are available, they have important drawbacks, including poor compliance due to frequent dosing requirements and side-effects[17]. Waning efficacy over time and the consequent need for treatment escalation are also observed, perhaps in part because the majority of medications in use do not address pathophysiological changes at the primary site of IOP regulation and aqueous humor egress from the eye, the trabecular meshwork (TM). These limitations result in a large proportion of glaucoma treatment regimens comprising more than one therapeutic agent and patients frequently changing medications or requiring treatment escalation that ultimately involves invasive surgery to maintain a clinically acceptable IOP[18]. Therefore, a substantial unmet need remains in the treatment of glaucoma to identify new therapeutic targets offering novel mechanisms of action, as well as treatment platforms that may offer increased durability and tolerability without compromising safety.

We performed genetic association analyses of IOP and glaucoma across eight cohorts to identify rare and coding variants that modulate the risk for glaucoma through IOP. This led to the identification of ANGPTL7 as a candidate, consistent with findings reported recently by another group[19]. In this report, we (1) strengthen the genetic link to glaucoma protection by (i) showing a consistent protective effect of the Gln175His variant in ANGPTL7 across eight cohorts, (ii) identifying a burden of ultra-rare missense variants in ANGPTL7 associated with reduced IOP levels, and (iii) identifying an additional rare ANGPTL7 loss-of-function variant in African-ancestry individuals; we also present (2) in vitro characterization of ANGPTL7 variants identified from genetic analyses; and (3) in vivo results showing that mice lacking Angptl7 have reduced basal IOP, and that even a partial knockdown of Angptl7 with small interfering RNA (siRNA) can lead to lowering of IOP in mice. Our results establish an important role for ANGPTL7 as a physiological regulator of IOP and suggest that it is also amenable to modification by pharmacological tools, making it a compelling target for a glaucoma therapeutic.

## Results

**Coding variants in ANGPTL7 are associated with reduced IOP.** We studied the effect of rare, protein-altering variation on IOP across two large cohorts, UK Biobank (UKB) and the Geisinger DiscovEHR (GHS), including 129,207 individuals of European descent after exclusion of cases with a glaucoma diagnosis (Methods, Supplementary Table 1). To increase our power to detect associations with rare variants, we performed burden tests by aggregating for each gene all rare (minor allele frequency (MAF) < 1%) predicted loss-of-function (pLOF, defined as stop-gain, frameshift, splice donor, splice acceptor, start-loss, and stop-loss) and missense (predicted deleterious by 5 algorithms, Supplementary Methods) variants. We observed a genome-wide significant association ($P < 5 \times 10^{-8}$) of variants in angiopoietin-like 7 (ANGPTL7) with reduced IOP ($\text{beta}_{\text{allelic}} = -0.21$ SD, $P = 5.3 \times 10^{-24}$; Fig. 1a). The gene burden included 63 rare variants but was dominated by two: a missense (Gln175His, MAF = 0.7%) and a stop-gain (Arg177*, MAF = 0.03%) variant, which accounted for 1902 and 82 individuals out of a total of 2188 carriers, respectively (Figs. 1b, 2, Supplementary Fig. 1). Exclusion of Gln175His and Arg177* from the burden meta-analysis between UKB and GHS did not eliminate the signal completely ($\text{beta}_{\text{allelic}} = -0.23$ SD, $P = 4.4 \times 10^{-4}$; Supplementary Fig. 2), suggesting that other ultra-rare variants in ANGPTL7 are also associated with reduced IOP.

In single-variant analyses, Gln175His was associated with reduced IOP at a genome-wide significant level ($\text{beta}_{\text{allelic}} = -0.20$ SD, $P = 3.1 \times 10^{-20}$, Fig. 2a). Heterozygous and homozygous carriers of Gln175His in ANGPTL7 had a 5.2% (0.8 mmHg) and 26.5% (4.1 mmHg) reduction in median IOP in UKB, respectively (Fig. 2b). The Arg177* variant was also nominally associated with reduced IOP with an effect size similar to that of Gln175His ($\text{beta}_{\text{allelic}} = -0.24$ SD, $P = 2.6 \times 10^{-2}$, Fig. 2c), and the 77 heterozygous Arg177* carriers had a 9% (1.4 mmHg) median IOP decrease (Fig. 2d). Arg177* appears to be the predominant pLOF variant in European populations; a burden test restricted to pLOF variants (15 variants across UKB and GHS) was dominated by Arg177* (82 of 112 total carriers) and was comparable ($\text{beta}_{\text{allelic}} = -0.21$ SD, $P = 2.2 \times 10^{-2}$; Supplementary Fig. 3) to the single-variant association of Arg177* with IOP.

We searched other ancestries for additional pLOFs in ANGPTL7 and identified Trp188*, which is enriched in individuals of African descent (MAF = 0.3%) compared to Europeans (MAF = 0.0013%). We performed an association of Trp188* with IOP in African ancestry individuals from UKB and the Primary Open Angle African American Glaucoma Genetics (POAAGG) study, followed by meta-analysis. Trp188* showed a trend towards reduced IOP, similar to Arg177* and Gln175His, but this was not statistically significant ($\text{beta}_{\text{allelic}} = -0.11$ SD, $P = 5 \times 10^{-1}$). A cross-ancestry meta-analysis of Arg177* and Trp188* variants showed a nominally significant association with reduced IOP ($\text{beta}_{\text{allelic}} = -0.21$ SD, $P = 1.5 \times 10^{-2}$; Supplementary Fig. 4).

In summary, we observed a significant association of Gln175His in ANGPTL7 with reduced IOP and a sub-threshold association, in the same direction and of similar magnitude, with pLOF variants in ANGPTL7. Assuming the pLOF variants indeed cause a loss of protein function, our data suggest that loss of ANGPTL7 can lead to lower IOP.

**IOP-associated variants in ANGPTL7 are protective against glaucoma.** To understand if carriers of variants in ANGPTL7 would also be protected against glaucoma, we performed an association analysis of Gln175His with glaucoma in UKB, GHS, and six additional studies: the Mount Sinai's BioMe Personalized

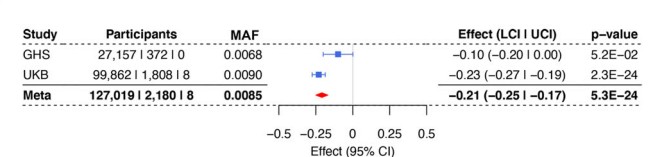

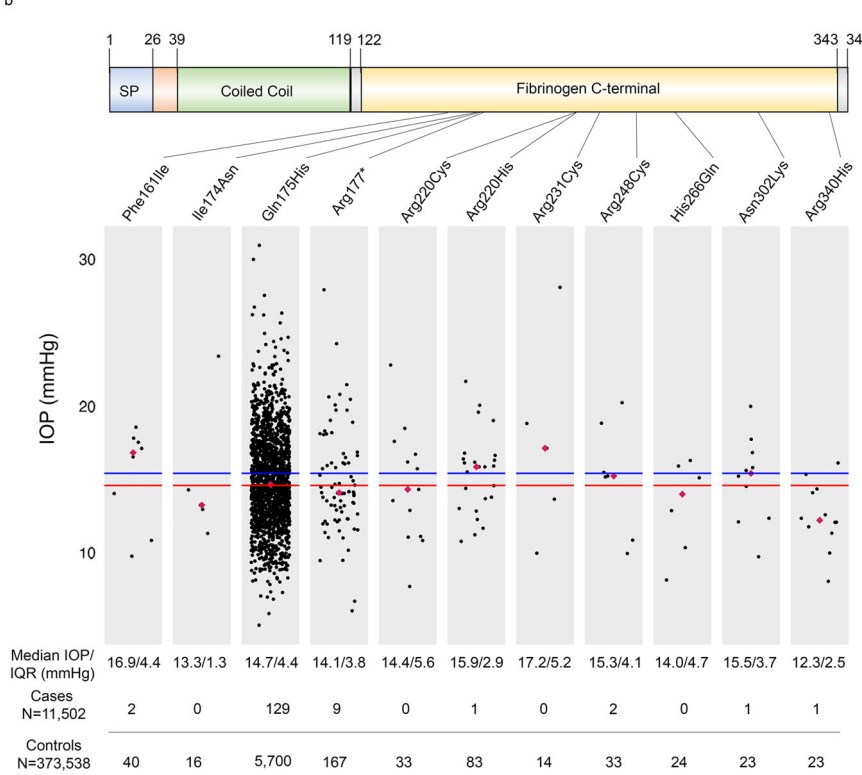

**Fig. 1 An aggregate of rare (MAF < 1%) loss-of-function and missense variants in _ANGPTL7_ is associated with IOP. a** Association of an aggregate of 63 pLOF and deleterious (based on 5 prediction algorithms) missense variants in _ANGPTL7_ with reduced IOP in 129,207 individuals of European descent. **b** Missense and predicted loss-of-function (pLOF) variants in _ANGPTL7_ and IOP levels in individuals of European descent from UKB. The plots represent Goldmann-correlated IOP (IOPg; mean of both eyes) levels in carriers of 1 pLOF and 10 missense variants in _ANGPTL7_ that are predicted deleterious by five different algorithms and have at least five carriers amongst the 101,678 exome-sequenced individuals with IOP measurements in the UK Biobank. The median IOP level across carriers of all 49 pLOF and predicted-deleterious missense _ANGPTL7_ variants (14.64 mmHg) is indicated by the red line, and the median IOP in non-variant carriers (15.46 mmHg) is indicated by the blue line. Magenta diamonds mark the median IOP in carriers of each variant. Beneath the plots is the median and interquartile range (IQR) of IOP and the numbers of variant carriers diagnosed with glaucoma or controls in UKB (n = 385,040). GHS Geisinger DiscovEHR, UKB UK Biobank, MAF Minor allele frequency.

Medicine Cohort from Mount Sinai Health System, New York (SINAI), the Malmö Diet and Cancer Study from Malmö, Sweden (MALMO), the FinnGen cohort from Finland, the Estonia Biobank at the University of Tartu, Estonia (EstBB), the HUNT study from Nord-Trøndelag, Norway (HUNT), and the Copenhagen General Population Study/Copenhagen City Heart Study from Copenhagen, Denmark (CGPS-CCHS). A meta-analysis across these eight cohorts showed a significant reduction in glaucoma risk for Gln175His carriers (odds ratio $(OR_{allelic}) = 0.77$, $P = 2.7 \times 10^{-6}$, Fig. 3a). We also analyzed glaucoma risk in carriers of the rarer Arg177*/Trp188* variants in a cross-ancestry meta-analysis and observed a consistent trend towards reduction in risk ($OR_{allelic} = 0.87$, $P = 4.1 \times 10^{-1}$, Fig. 3b). Taken together, the associations of missense and pLOF variants in _ANGPTL7_ with reduced IOP and the association of the missense variant with reduced glaucoma risk suggest the hypothesis that loss of ANGPTL7 confers protection against glaucoma, and that this effect is mediated through the regulation of IOP.

**_ANGPTL7_ variants are associated with corneal measures**. We performed a phenome-wide association analysis (PheWAS) to understand whether other traits were associated with a burden of pLOF and deleterious missense variants in _ANGPTL7_. We tested the _ANGPTL7_ variant aggregate for association with 14,050 and 10,032 binary and quantitative traits in UKB and GHS, respectively. No associations reached phenome-wide significance ($P < 2 \times 10^{-6}$ after multiple-testing correction for 24,082 total traits) in GHS. The only significant associations in UKB were with ocular traits (Table 1), specifically with decreased corneal-compensated IOP (IOPcc), decreased corneal resistance factor (CRF) and increased corneal refractive power along both weak and strong meridians measured at 3 and 6 mm diameters. The effect of these variants on IOPcc was slightly attenuated (−0.17 SD) compared to that on IOPg (−0.23 SD in UKB), which suggests that ANGPTL7 has some impact on corneal properties that are known to affect the IOPg measurements[20]. The association observed with decreased CRF is also consistent with a corneal effect of ANGPTL7.

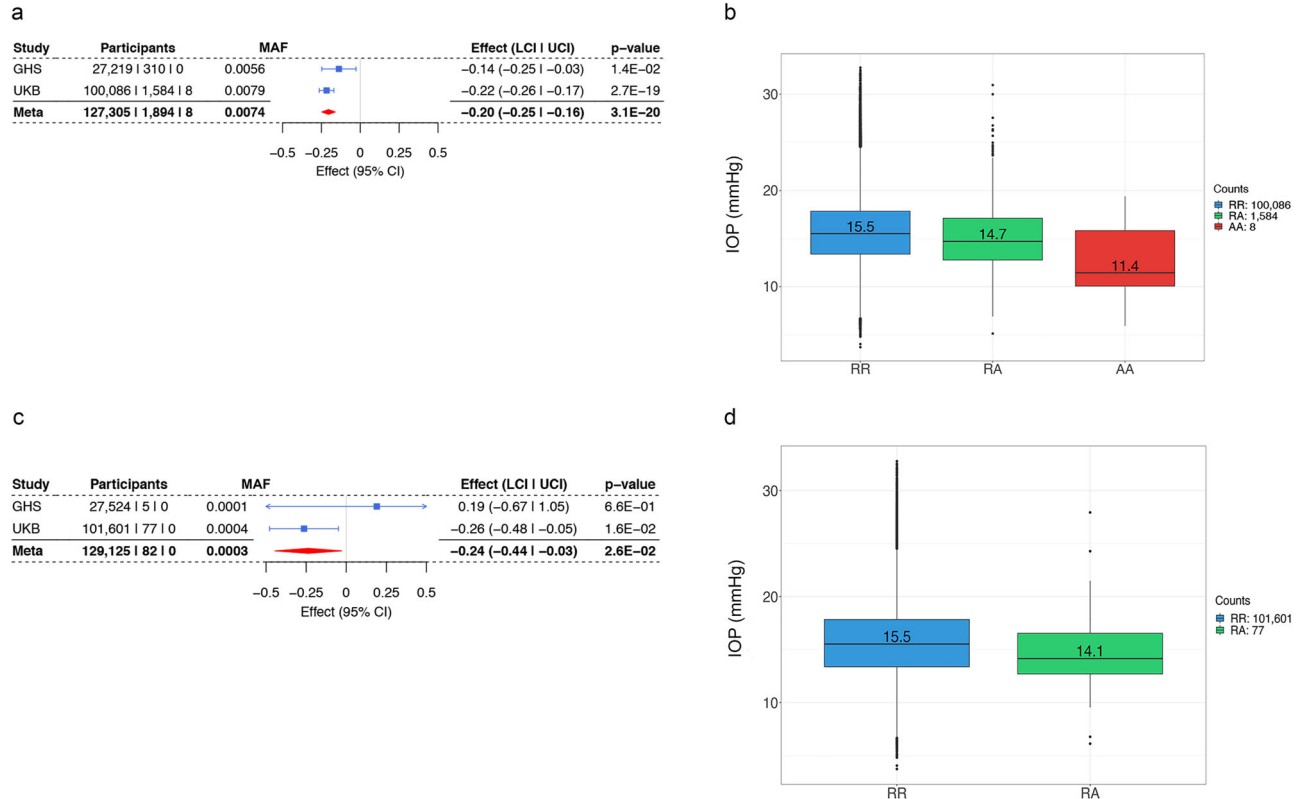

**Fig. 2 Gln175His and Arg177\* are major contributors to the gene burden association of _ANGPTL7_ with IOP.** Association of Gln175His (**a**) and Arg177\* (**c**) variants in _ANGPTL7_ with IOP, effect measured in standard deviation units, in individuals of European descent. **b**, **d** Boxplots representing IOPg in the UK Biobank across genotypes. **b** Gln175His heterozygous and homozygous carriers have a 0.8-mmHg and 4.1-mmHg lower median IOPg, respectively, compared to non-carriers. **d** Arg177\* heterozygous carriers have a 1.4-mmHg lower IOPg compared to non-carriers. GHS Geisinger DiscovEHR, UKB UK Biobank, MAF Minor allele frequency.

We used the autorefraction measurements at 3 mm diameter to derive measures of clinical interest, namely, mean corneal refractive power (mCRP), corneal astigmatism, and refractive astigmatism (Supplementary Methods) and checked for association with _ANGPTL7_. We observed a significant association with increased mCRP (beta$_{allelic}$ = 0.16 SD, $P = 1.1 \times 10^{-13}$, Table 1) but no association with corneal or refractive astigmatism. We also did not observe associations with mean spherical equivalent (MSE; measure of refractive error) or myopia (either derived from MSE or via ICD-10 diagnosis), which could result from increased mCRP (Supplementary Table 2). Overall, our PheWAS results show that while _ANGPTL7_ is associated with changes in corneal anatomy/biomechanics-related quantitative measures, we did not detect an increased risk for any related disease outcomes that we could test. In addition, pLOF and deleterious missense variants in _ANGPTL7_ are not associated with any systemic quantitative traits or binary outcomes.

**Gln175His, Arg177\* and Trp188\* are defective in secretion**. To understand the impact of Gln175His, Arg177\*, and Trp188\* variants on the expression and secretion of ANGPTL7, we transiently transfected constructs expressing the human wild type (WT), Gln175His, Arg177\*, and Trp188\* proteins in HEK293 cells. We measured mRNA levels by Taqman, which showed similar Gln175His and Arg177\* transcript levels and a trend towards decreased levels of the Trp188\* transcript compared to WT ($P = 0.08$; Supplementary Fig. 5). However, analysis of intracellular, steady-state protein in whole-cell lysate by western blotting and ELISA revealed increased levels of Gln175His compared to WT ($P < 1 \times 10^{-4}$; Fig. 4c). As expected, Arg177\*

and Trp188\* encoded lower molecular weight proteins (~30-32 kDa). No significant difference in the protein levels of these two mutants was revealed by ELISA in comparison to WT (Fig. 4a, c). Because ANGPTL7 is a secreted protein, we next determined the levels of WT, Gln175His, Arg177\*, and Trp188\* in the cellular supernatant. Protein analysis by western blot showed that the Arg177\* and Trp188\* variants were not detectable and the Gln175His was drastically reduced in the supernatant compared to WT ($P < 0.05$; Fig. 4b). ELISA assay further corroborated the severely reduced levels of Gln175His and the inability of Arg177\* and Trp188\* to reach the extracellular space (Fig. 4c, d).

**ANGPTL7 is expressed in cornea, TM and sclera across species**. To identify expression of _ANGPTL7_ in ocular tissues across different species, transcriptome profiles from different parts of the eye were generated (Supplementary Methods). High _ANGPTL7_ expression was observed in cornea, TM, and sclera in human and African green monkey eyes (Fig. 5a, b, Supplementary Fig. 6). High _Angptl7_ expression was also observed in cornea, TM, sclera, optic nerve head, and choroid/RPE in eyes of C57BL/6 J mice (Fig. 5c). In situ hybridization on human donor and mouse eyes using RNAscope probes for human _ANGPTL7_ and mouse _Angptl7_ showed _ANGPTL7/Angptl7_ expression in TM, cornea stroma, and sclera (Fig. 5d, e; Supplementary Methods).

**Increasing levels of mAngptl7 in mouse eyes increases IOP.** Previous studies showed that overexpression of _ANGPTL7_ in TM cells leads to changes in extracellular matrix (ECM) deposition

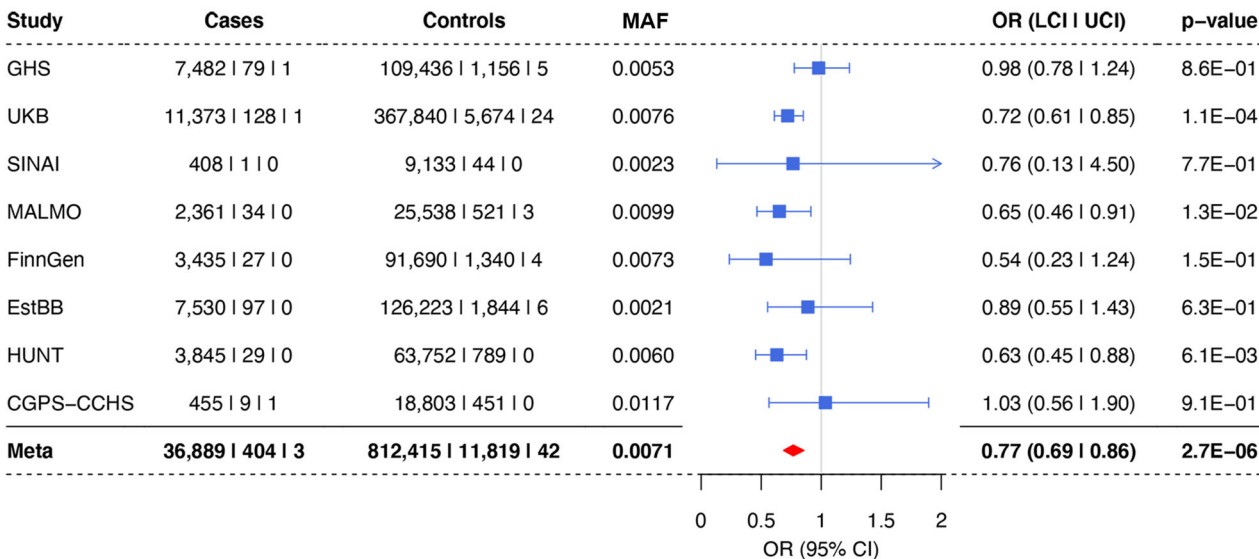

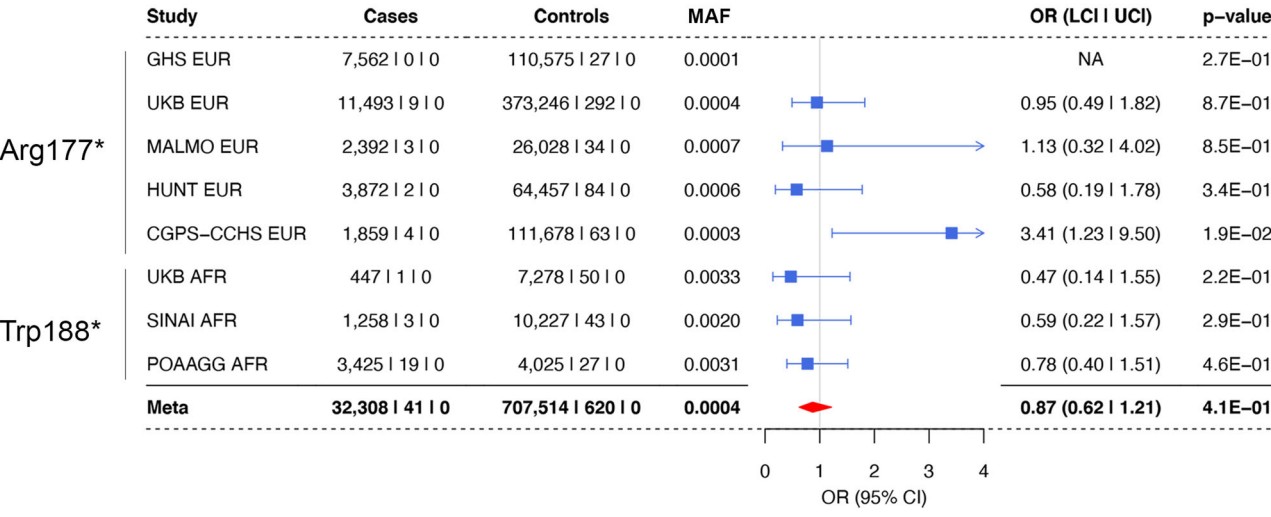

**Fig. 3 Association of _ANGPTL7_ variants with glaucoma. a** Meta-analysis results for Gln175His with glaucoma across 8 different cohorts. **b** Cross-ancestry meta-analysis of Arg177* and Trp188* across 5 European (EUR) and 3 African (AFR) ancestry cohorts. The variants in the meta-analysis of EUR and AFR cohorts were Arg177* and Trp188*, respectively. GHS Geisinger DiscovEHR, UKB UK Biobank, SINAI Mt. Sinai Medical School BioMe Biobank, MALMO Malmö Diet and Cancer Study, EstBB the Estonia Biobank at the University of Tartu, HUNT the HUNT study from Nord-Trøndelag, CGPS-CCHS the Copenhagen General Population Study and the Copenhagen City Heart Study, POAAGG Primary Open Angle African-American Glaucoma Genetics, MAF Minor allele frequency.

and reorganization[21,22] and that ANGPTL7 is increased in aqueous humor of glaucoma patients[22], however, the role of ANGPTL7 in IOP regulation is not clear. To investigate this, we injected mAngptl7 protein in mice via intravitreal and intracameral routes and measured IOP over time. Intravitreal injection of murine Angptl7 (mAngptl7) in mice led to an initial drop in IOP followed by, starting on day 4, an elevation in IOP of 4–5 mmHg, a 22–25% increase compared to baseline, that lasted until the end of the experiment on day 7 (Fig. 6a). Similarly, intracameral injection of mAngptl7 in mice led to an initial drop and subsequent elevation (by 2–5 mmHg) of IOP, starting on day 3 until the end of the experiment on day 7 (Fig. 6b). Vehicle-

injected mice did not show an increase in IOP in either route of administration.

**_Angptl7_ KO mice have lower basal IOP than WT.** We generated and characterized _Angptl7_−/− (KO) mice. We confirmed via RNAscope that _Angptl7_ mRNA was not expressed in any ocular tissue in KO mice whereas it was expressed in TM, cornea, and sclera of WT mice (Fig. 7; Supplementary Methods). In addition, histological analysis of the eye showed no difference in the ocular angle in KO mice compared with WT (Supplementary Fig. 7). We did not observe any ocular changes on anterior segment optical

**Table 1 Statistically significant ($P < 2 \times 10^{-6}$) results from PheWAS of an aggregate of up to 110 pLOF and deleterious missense variants (MAF < 1%) in *ANGPTL7* in UKB.**

| Trait | P value | OR/Effect in SD (LCI \| UCI) |
|---|---|---|
| Corneal-ompensated intraocular pressure (IOPcc) (mean of both eyes) | 6.32E-14 | −0.17 [−0.21 \| −0.13] |
| CRF (right eye) | 5.20E-13 | −0.16 [−0.20 \| −0.11] |
| CRF (left eye) | 4.50E-12 | −0.15 [−0.20 \| −0.11] |
| 6 mm weak meridian (left eye) | 4.00E-20 | 0.21 [0.16 \| 0.25] |
| 6 mm weak meridian (right eye) | 4.20E-18 | 0.19 [0.15 \| 0.24] |
| 6 mm strong meridian (left eye) | 3.10E-17 | 0.19 [0.14 \| 0.23] |
| 6 mm strong meridian (right eye) | 7.60E-17 | 0.18 [0.14 \| 0.23] |
| 3 mm weak meridian (right eye) | 2.00E-14 | 0.16 [0.12 \| 0.20] |
| 3 mm weak meridian (left eye) | 1.60E-13 | 0.15 [0.11 \| 0.20] |
| 3 mm strong meridian (left eye) | 1.30E-12 | 0.15 [0.11 \| 0.19] |
| 3 mm strong meridian (right eye) | 1.80E-12 | 0.15 [0.11 \| 0.19] |
| Corneal power (mean of both eyes) | 1.10E-13 | 0.16 [0.11 \| 0.20] |

*CRF* Corneal resistance factor.

coherence tomography, or a difference in corneal thickness between the two genotypes (Supplementary Fig. 8). We also monitored the IOP in KO, *Angptl*$^{+/-}$ (Het) and WT mice and observed a dose-dependent decrease in IOP across the three genotypes (Fig. 8a). The mean IOP was lowered in KO mice (mean ± standard error of the mean (SEM): 15.39 ± 0.25 mmHg) by 11% (1.96 mmHg, $P < 1 \times 10^{-4}$) compared to WT (17.36 ± 0.23 mmHg). Het mice (16.26 ± 0.43 mmHg) showed a smaller (6%, 1.1 mmHg, $P = 0.02$) but significant reduction in IOP compared to WT.

***Angptl7* KO mice have increased conventional outflow facility**. *Angptl7* is highly expressed in cornea and TM and since we used a Tonolab rebound tonometer to measure IOP, we wanted to further confirm that Angptl7 function in the cornea was not affecting IOP. Therefore, we measured the conventional outflow facility using constant flow infusion to understand the relationship between IOP and outflow resistance in *Angptl7* KO ($n = 7$) and WT mice ($n = 4$). The conventional outflow facility was increased by 21% in KO (25.18 nL/minute/mmHg) compared to WT mice (20.85 nL/minute/mmHg; $P = 0.15$; Fig. 8b). The mean increase in outflow facility was consistent with the mean IOP lowering in KO, according to the modified Goldman equation.

**siRNA-induced knockdown of *Angptl7* mRNA and lowering of IOP in WT Mice**. To investigate whether knockdown of *Angptl7* with siRNA can also lower IOP, we tested six different siRNAs (Table 2) targeting *Angptl7* in C57BL/6 J mice and monitored IOP over time. We injected C57BL/6 J mice intravitreally with 15 µg of siRNAs and performed qPCR six weeks later on limbal rings dissected from mouse eyes enriched for the TM. IOP was significantly lowered two weeks post-injection in mice treated with two of the six siRNAs (siRNA #3 and #5) compared to the PBS and Naïve (no injection) groups (Fig. 8c, Supplementary Data 1). Naïve and PBS-treated animals maintained their IOPs at baseline for the duration of the study (weeks 0–6). In mice treated with siRNA#3 and #5, IOP was lowered by 2–4 mmHg starting at week 2 compared to PBS-treated mice (Fig. 8c). At the end of the study, we collected the eyes, carefully micro-dissected the limbal ring and performed qPCR. We observed the highest level of knockdown (>50%) of *Angptl7* mRNA with siRNAs #3 and #5 compared to PBS-treated mice (Fig. 8d), which is consistent with

the IOP lowering observed in mice injected with these two siR-NAs. These results suggest that acute inhibition of *Angptl7* expression also lowers IOP.

**Discussion**

In this study, we present genetic and functional evidence for a role for ANGPTL7 in the physiological control of IOP and as a potential target for glaucoma therapy. A recent publication by Tanigawa et al.[19] identified a protective association of *ANGPTL7* with glaucoma and in this paper we further evaluate and strengthen this finding by (1) adding six cohorts to the genetic association analyses in UK Biobank and FinnGen that were used in Tanigawa et al, and showing a consistent protective trend in the majority of these for the rare missense variant, Gln175His (rs28991009); (2) identifying an African ancestry-enriched pLOF variant, Trp188*, in *ANGPTL7* in addition to Arg177* in Europeans and showing that both variants trend towards protection from glaucoma (3) identifying through exome sequencing and gene burden analyses multiple predicted-deleterious *ANGPTL7* variants associated with reduced IOP, indicating that there may be other ultra-rare *ANGPTL7* variants that confer protection from glaucoma. Together, these new analyses provide further support for the role of ANGPTL7 in IOP regulation and glaucoma pathogenesis. We also show through cell-based expression assays, that Gln175His, Arg177*, and Trp188* were severely defective in secretion when compared to wild-type and, while not proof, this observation is consistent with the hypothesis that the three variants result in a loss of protein function.

*ANGPTL7* is a secreted glycoprotein that is thought to form homo-oligomers (tri- or tetramers), like other members of the ANGPTL family[23,24]. The *ANGPTL7* mRNA was first isolated from human postmortem eyes where it shows the highest expression relative to other tissues[23]. Within the eye, our in situ hybridization results show that *ANGPTL7* is expressed most strongly in the cornea and TM, consistent with previous findings[22]. We have also shown using single-cell RNAseq that *ANGPTL7* expression is particularly enriched in the juxtacanalicular tissue (JCT), a region of the TM most important for IOP regulation and generation of aqueous humor outflow resistance[25,26]. In addition to being expressed in tissues directly relevant in glaucoma, several lines of evidence have implicated ANGPTL7 in glaucoma pathophysiology. First, elevated levels of *ANGPTL7* mRNA and protein were observed in eye tissues from glaucoma patients compared to controls, and under conditions of increased IOP simulated by perfusion of eye anterior segment explants[22,27,28]. Second, *ANGPTL7* is one of the most upregulated genes in response to corticosteroid treatment[29,30], which can cause increased IOP in ~40% of general population and ~90% of individuals with POAG[31,32]. Third, *ANGPTL7* levels are also increased in response to TGF-beta, a growth factor that is thought to modulate the ECM and lead to increased IOP[33]. Fourth, ANGPTL7 itself can modulate the expression of components of the TM ECM[21,22].

The above studies suggest a strong correlation of ANGPTL7 expression with glaucoma disease-state, however, they are not evidence for a causal relationship between ANGPTL7 levels and elevated IOP/glaucoma. Human genetics finding of an association between ANGPTL7 loss and lower IOP suggests that ANGPTL7 is physiologically important for IOP regulation. We validated this hypothesis in mice where we performed reciprocal experiments: measuring IOP after increasing ANGPTL7 levels via injection of mAngptl7 into mouse eyes, and after removing all mAngptl7 protein by generating *Angtpl7* KO mice. Our findings show that increasing mAngptl7 results in increased (~2–4 mmHg) IOP and decreasing mAngptl7 through KO mice

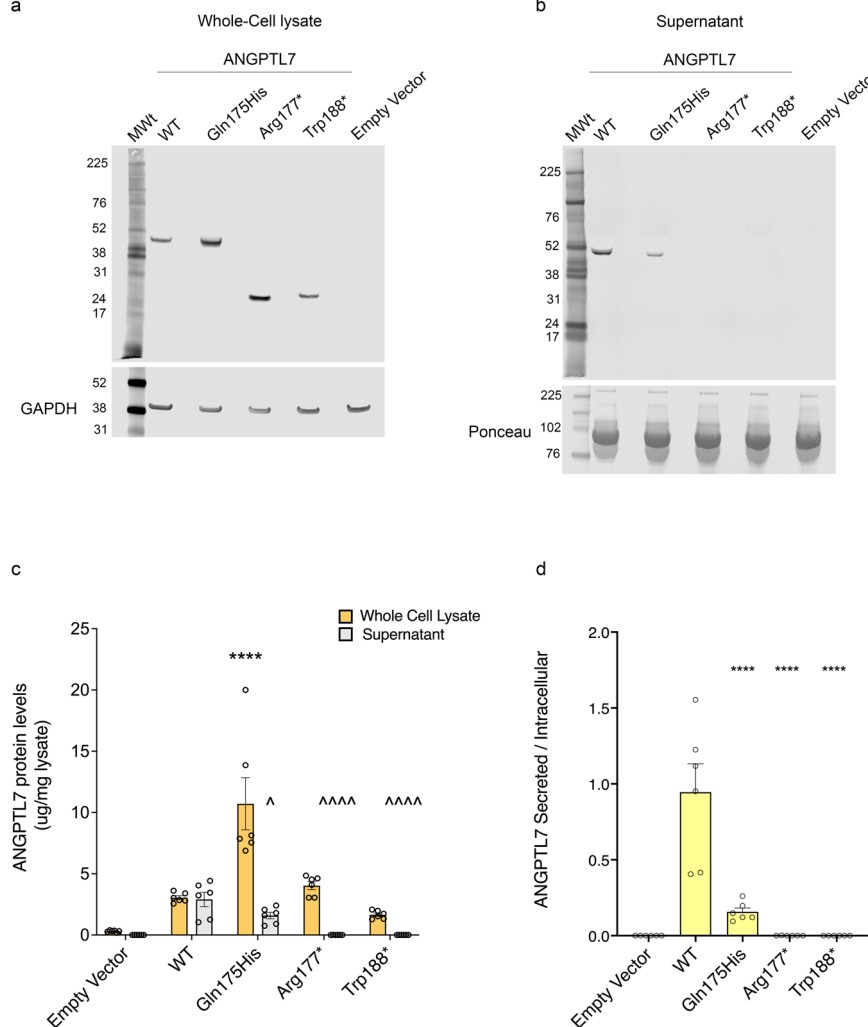

**Fig. 4 Expression analysis of ANGPTL7 Gln175His, Arg177*, and Trp188* in a HEK293 cell line. a**, **b** Western blotting shows intra- and extra-cellular protein levels of ANGPTL7 wild-type (WT), Gln175His, Arg177*, and Trp188*. **c** ELISA was run to quantify intra- and extra-cellular protein levels of ANGPTL7 WT, Gln175His, Arg177*, and Trp188* transiently transfected in HEK293 cells (whole-cell lysate was diluted 1:1,000; supernatant was diluted 1:10,000. Both whole-cell lysate and supernatant were normalized against the total amount of protein from the whole-cell lysate); $****= P < 1 \times 10^{-4}$ statistical difference from WT whole cell lysate; $\hat{} = P < 0.05$, $\hat{}\hat{}\hat{} = P < 1 \times 10^{-4}$ statistical difference from WT supernatant. **d** Ratio of secreted versus intracellular ANGPTL7 WT, Gln175His, Arg177*, and Trp188* protein levels. Raw ANGPTL7 WT, Gln175His, Arg177*, and Trp188* protein levels were normalized to the whole-lysate protein concentration; $**** = P < 1 \times 10^{-4}$ statistical difference from WT. Western blotting and ELISA analysis were repeated on three independent biological replicates. Technical replicates ($n = 3$) were run for the ELISA analysis. $P$ values were calculated by one-way ANOVA with Tukey's post hoc analysis. All data are presented as mean and error bars indicate the standard error of the mean (SEM). MWt molecular weight marker.

reduces basal IOP levels (~2 mmHg). We also observed a trend towards increased conventional outflow facility (~21%) in KO compared to WT mice that was consistent with the decrease in IOP in KO mice. Together, these findings suggest that ANGPTL7 functions in vivo to maintain IOP homeostasis. We further recapitulated the reduction in IOP observed in KO mice by injecting WT mouse eyes with siRNA against *Angptl7*, which not only replicates the observation in genetic mutant mice but also illustrates that the effect of mAngptl7 on IOP continues post-development and is amenable to modulation by therapeutics in adulthood. Our results of lower IOP in *Angptl7* KO mice are highly consistent with observations from human genetics. Based on this, we could extend the siRNA knockdown findings to humans and surmise that inhibition of ANGPTL7 in adulthood could be an efficacious strategy to lower IOP and, eventually, the risk for glaucoma.

Our human genetics findings indicate that the variants in *ANGPTL7* associated with reduced IOP were also associated

with decreased mean corneal resistance factor (CRF), an output of the Ocular Response Analyzer that reflects the elastic property of the cornea[34], and increased mean corneal refractive power (mCRP). mCRP, along with axial length and lens power, contributes to the overall refractive state of the eye[35], and increased mCRP (i.e. a steeper cornea) may be indicative of possible myopia or astigmatism. However, as we did not observe evidence of association with myopia or astigmatism in PheWAS, we conclude that the changes to mCRP are either insufficient to result in a disease outcome or are compensated by other changes in traits that were not measured, such as axial length[36]. While an effect of ANGPTL7 on corneal properties is clearly indicated by the associations with mCRP and CRF, we assert that this effect cannot account for the full extent of IOP reduction observed. Instead, we posit an independent contribution of ANGPTL7 to IOP homeostasis based on the following: (1) The persistent association with reduced IOPcc suggests that there is IOP reduction even after controlling for

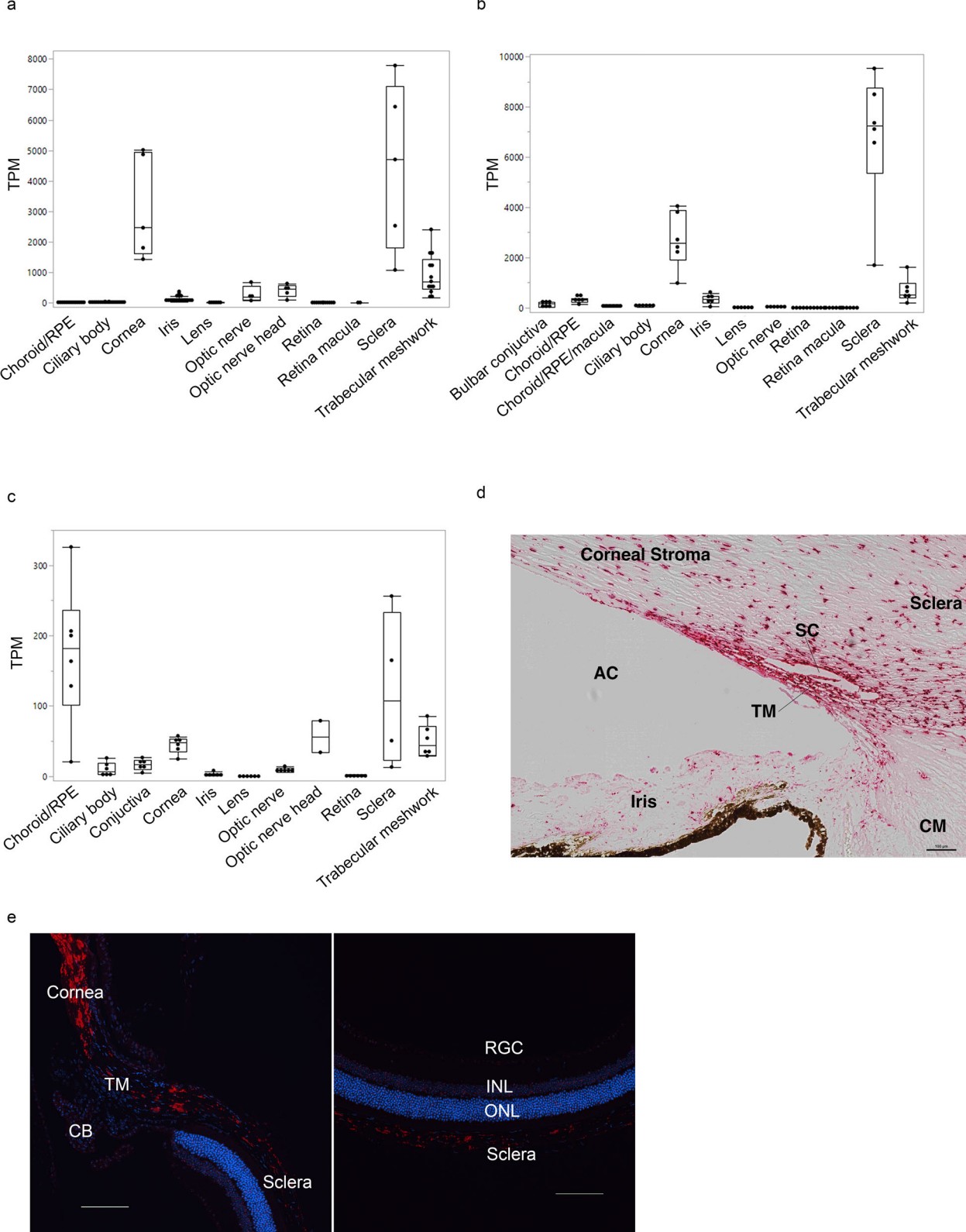

**Fig. 5 ANGPTL7 expression in ocular tissues across species.** RNA-sequencing-based expression levels (measured in transcripts per million, TPM, and represented as median and interquartile range) are highest in cornea, trabecular meshwork (TM), and sclera in human (**a**), and African green monkey (**b**) eyes, and in cornea, TM, sclera, optic nerve head, and choroid/RPE in C57BL/6 J mouse eyes (**c**); In situ hybridization (RNAscope) shows *ANGPTL7/* Angptl7 (red) expression in TM, cornea and sclera in human (**d**) and murine (**e**) eyes. Scale bars represent 100 μm. DAPI staining (blue) counterstains cell nuclei. RPE retinal pigmented epithelium, CB ciliary body, SC Schlemm's canal, CM ciliary muscle, AC anterior chamber, RGC retinal ganglion cell, INL inner nuclear layer, ONL outer nuclear layer.

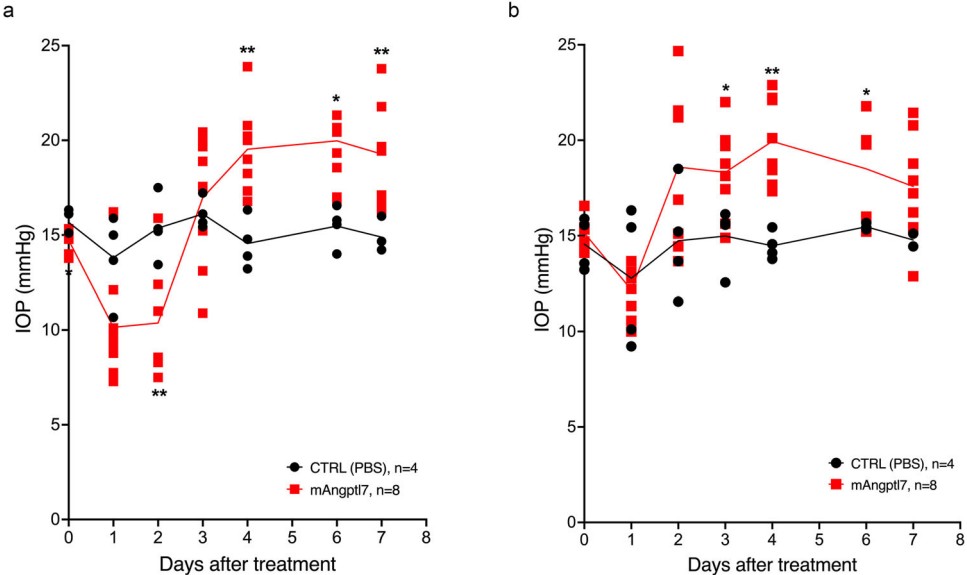

**Fig. 6 Increasing mAngptl7 levels in mouse eyes increases IOP.** Murine Angptl7 (mAngptl7) protein was injected into mouse eyes via intravitreal (**a**) or intracameral route (**b**), and IOP was measured over time. After an initial drop, IOP remained elevated for several days in mAngptl7-treated eyes compared to PBS treated (CTRL (PBS)) eyes. * = $P < 0.05$; ** = $P < 0.01$ statistical significance compared to PBS treatment. Statistical analyses were performed using Student's *t* test.

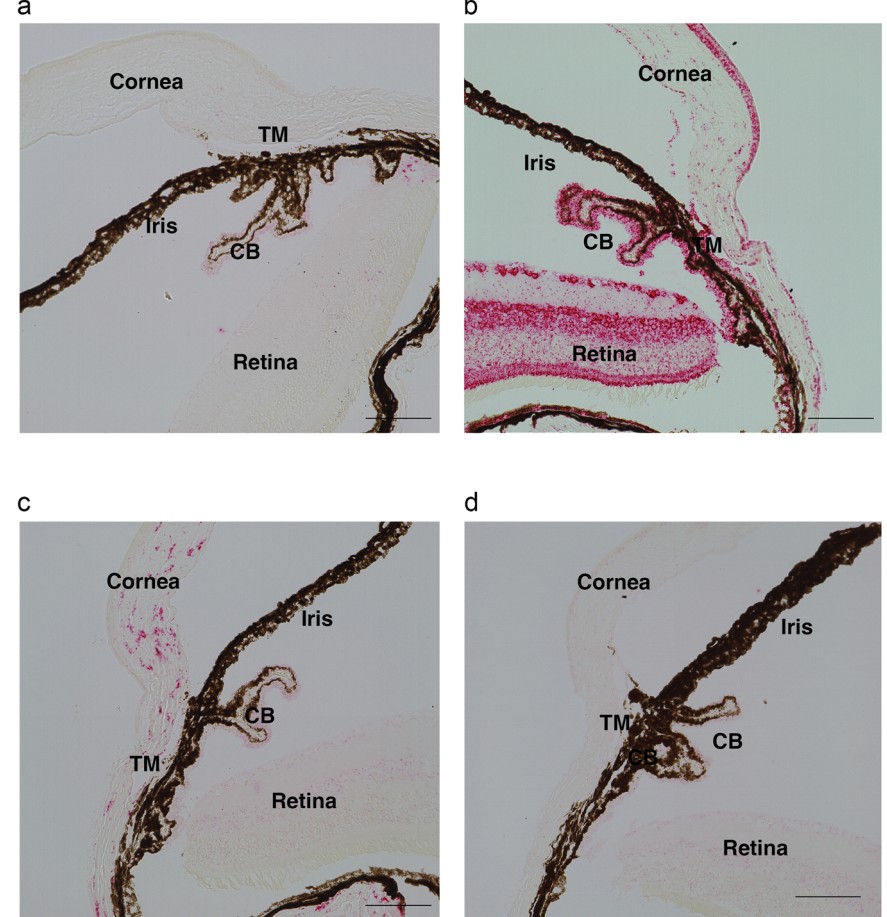

**Fig. 7 In situ characterization of Angptl7 mRNA in WT and *Angptl7* KO mouse eyes.** Angptl7 mRNA was not expressed in any ocular tissue in *Angptl7* KO mice whereas it was expressed in TM, cornea, and sclera of WT mice as shown by in situ hybridization (RNAscope). Brightfield images showing following probes. **a** Negative control: DapB. **b** Positive control: Ubc (red). **c** WT mice: Angptl7 (red), and (**d**) *Angptl7* KO mice: Angptl7 (no signal). Scale bars represent 100 μm.

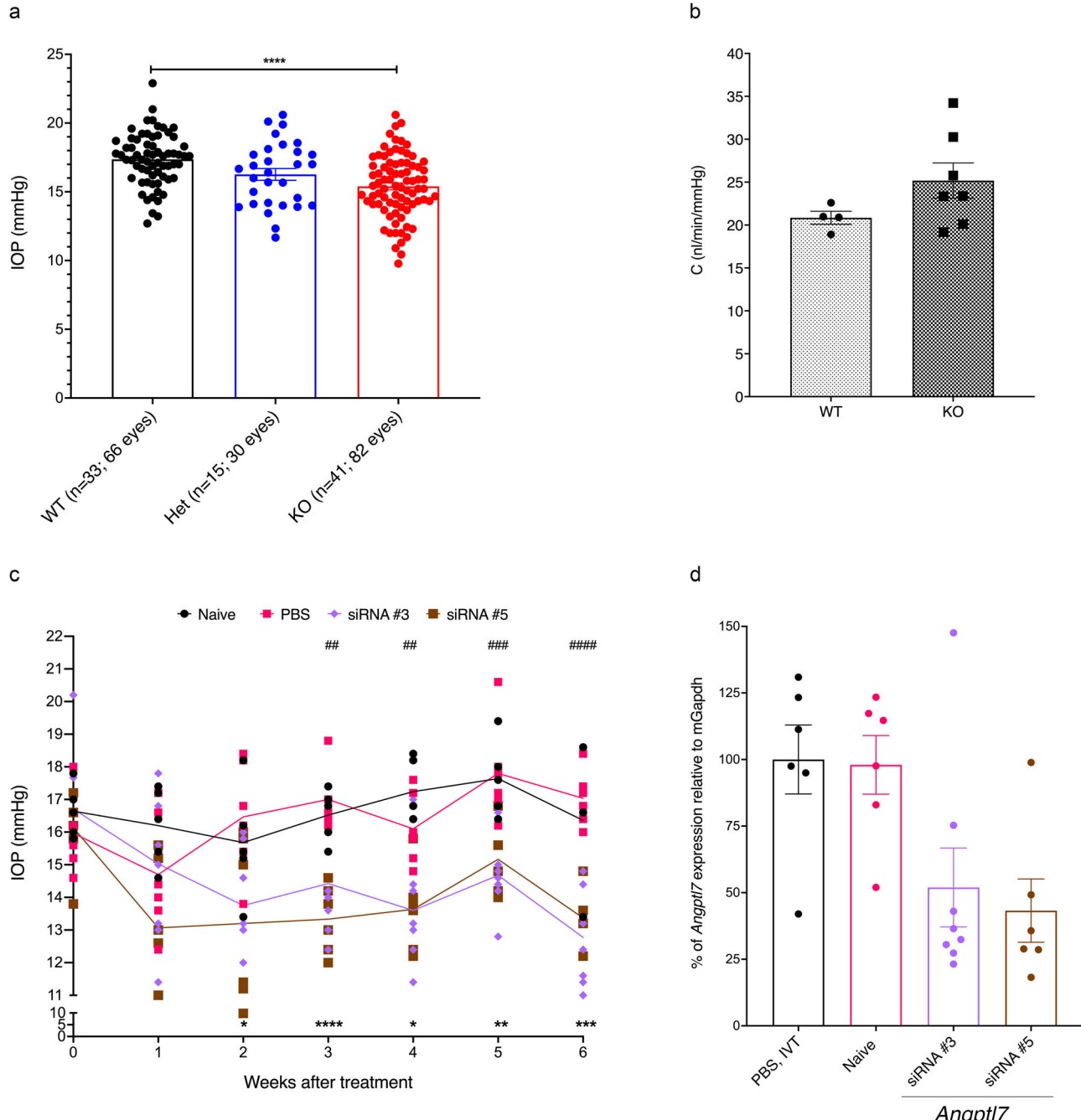

**Fig. 8 Reducing mAngptl7 levels in mice lowers IOP. a** IOP was significantly lowered in *Angptl7* KO compared to Het and WT mice (mean ± SEM; **** = $P < 1 \times 10^{-4}$). **b** Comparison of conventional outflow facility (C) between KO and WT mice. In KO mice ($n = 7$), C was increased compared to WT mice ($n = 4$, $P = 0.16$, unpaired Student's *t* test). Data are presented as mean ± SEM. **c** Intravitreal injection with 15 μg of Angptl7-siRNA significantly lowered IOP in two of six siRNAs tested ($n = 6$–8/group) compared to the PBS ($n = 6$) and Naïve (no injection, $n = 5$) groups and remained lowered throughout the end of the study. siRNAs 3 and 5 lowered IOP between 2 and 4 mmHg starting at week 2 compared to PBS-treated mice. * = $P < 0.05$, ** = $P < 0.01$, *** = $P < 1 \times 10^{-3}$, **** = $P < 1 \times 10^{-4}$ statistical difference between the mean IOP for siRNA #5 and PBS-treatment.# = $P < 0.05$,## = $P < 0.01$, ### = $P < 1 \times 10^{-3}$, #### = $P < 1 \times 10^{-4}$ statistical difference between the mean IOP for siRNA #3 and PBS-treatment. **d** qPCR results from micro-dissected limbal ring showed the highest level of knockdown (>50%) of Angptl7 mRNA with siRNAs #3 and #5 compared to PBS-treated mice, which is highly consistent with the IOP lowering observed in mice injected with these two siRNAs (**b**). Data are presented as mean and error bars represent SEM.

corneal properties; (2) *Angptl7* KO mice eyes show reduced basal IOP without evidence of corneal thinning or other corneal abnormalities; and (3) the association of *ANGPTL7* variants with glaucoma protection is a result we would not expect if the reduction of IOP was purely due to ANGPTL7's effect on

corneal anatomy[37]. Therefore, we believe that ANGPTL7 likely has a pleiotropic effect on both IOP and corneal anatomy/ biomechanics in humans.

Therapeutic knockdown of ANGPTL7 in patients with open angle glaucoma may present a unique opportunity to not only

**Table 2 Mouse Angptl7 siRNA sequences used to knockdown mAngptl7.**

| siRNA | sense strand | antisense strand |
|---|---|---|
| #1 | UUGGGCAAUGAACUGAACAGA | UCUGUUCAGUUCAUUGCCCAACG |
| #2 | GUACCAGAAGAACUACCGAAA | UUUCGGUAGUUCUUCUGGUACAG |
| #3 | AGACAGUAUAAGCAAGGGUUA | UAACCCUUGCUUAUACUGUCUCC |
| #4 | GCAGAAGCCUCAUAAACGCAA | UUGCGUUUAUGAGGCUUCUGCAG |
| #5 | ACACUUCCUUGUGUCUAUAGA | UCUAUAGACACAAGGAAGUGUCG |
| #6 | CUGCAGAAGCCUCAUAAACGA | UCGUUUAUGAGGCUUCUGCAGCC |

lower IOP through a novel mechanism of action, but also to intervene in the disease process. Among the most widely used IOP-lowering agents, none directly address pathophysiological changes occurring at the level of TM and instead reduce IOP through mechanisms that reduce the aqueous burden on the conventional pathway, either through suppressing aqueous formation or increasing its outflow via the alternative (uveoscleral) pathway[38]. Consequently, as TM dysfunction progresses in a glaucoma patient, treatment intensification naturally follows[18]. While further studies are needed to investigate the role of ANGPTL7 in pathways contributing to TM dysfunction and IOP elevation, several lines of evidence point to pro-fibrotic actions at the level of the TM, including its corticosteroid and TGF-beta responsiveness, as well as its enriched expression in JCT TM (discussed earlier).

As all studies, this study has limitations. First, we did not account for IOP-lowering medication use in the IOP analysis. We did exclude from the IOP analysis individuals with a glaucoma diagnosis, greatly reducing the effect of medication use, however this would not exclude those individuals on IOP-lowering medications without a glaucoma diagnosis. In addition, the IOP measurement in UKB for most participants was only taken once, and therefore can be prone to errors that a median of multiple measurements might buffer against. Both of the above factors would influence the estimation of the effect of rare genetic variants on IOP. Second, since a glaucoma diagnosis can often be made well after the onset of disease, we can expect that a certain percentage of controls have undiagnosed glaucoma, which may affect the effect size estimate. Third, the ANGPTL7 gene in humans lies within intron 28 of MTOR[39], therefore variants in ANGPTL7 could also affect MTOR function. While a possibility, it is unlikely that the effect on IOP and glaucoma is due to MTOR, as: (i) the variants are predicted to be protein-altering in ANGPTL7 whereas there is no obvious predicted functional effect on MTOR; (ii) we show data suggesting that the variants have a functional impact on ANGPTL7; and (iii) we show mouse data that establish a role for ANGPTL7 in IOP regulation.

In summary, our genetic and pharmacological results indicate that ANGPTL7 participates in the normal physiological regulation of IOP in humans and mice. Since excessive amounts of mAngptl7 protein in the eyes of experimental animals cause IOP to elevate to pathological levels, upregulation of ANGPTL7 in humans may be responsible for the elevated IOP that leads to POAG. Therefore, we propose ANGPTL7 as an excellent candidate to explore as a therapeutic target for POAG.

## Methods

**Ethics approval and informed consent**. All participants provided informed consent, and studies were approved by the individual IRBs at the respective institutions. UK Biobank has approval from the North West Multi-center Research Ethics Committee (MREC), which covers the UK. It also sought the approval in England and Wales from the Patient Information Advisory Group (PIAG) for gaining access to information that would allow it to invite people to participate. The DiscovEHR study was approved by the Institutional Review Board (IRB) at Geisinger. The BioMe Biobank is an ongoing research biorepository approved by the Icahn School of Medicine at Mount Sinai's IRB. The

Ethical Committee at Lund University approved the Malmo Diet and Cancer Study (LU 51–90) and all the participants provided a written informed consent. The CGPS study (H-KF-01-144/01) was approved by the Ethics Committee of the Capital Region and from the Danish Data Protection Agency. Research at Estonian Biobank is regulated by Human Gene Research Act and all participants have signed a broad informed consent. IRB approval for current study was granted by Research Ethics Committee of University of Tartu, approval nr 236/T-23. For the POAAGG study, approval to enroll and to recontact subjects was obtained from the University of Pennsylvania IRB. The Finngen Biobank was evaluated and approved by the Coordinating Ethics Committee of the Helsinki and Uusimaa Hospital District.

**Study design**. Association with IOP was tested on a total of 101,678 individuals and 27,529 individuals of European ancestry from the United Kingdom Biobank (UKB) and the MyCode Community Health Initiative cohort from Geisinger Health System (GHS), respectively. The UKB is a population-based cohort study of people aged between 40 and 69 years recruited through 22 testing centers in the UK between 2006 and 2010[40]. The GHS MyCode study is a health system-based cohort of patients from Central and Eastern Pennsylvania (USA) recruited in 2007–2019[41]. For IOP association tests in African ancestry individuals, we included 4114 individuals from UKB and 3167 individuals from the Primary Open Angle African-American Glaucoma Genetics (POAAGG) study conducted at the University of Pennsylvania Perelman School of Medicine[42]. We excluded all participants with a glaucoma diagnosis code (ICD-10 H40) or self-reported glaucoma (UKB field IDs: 6148 and 20002) from IOP analyses.

Association of ANGPTL7 variants with glaucoma was tested in 8 studies: UKB, GHS, Mt. Sinai BioMe cohort (SINAI), the Malmö Diet and Cancer study (MALMO)[43], the Estonia Biobank (EstBB)[44], The Trøndelag Heath Study (HUNT)[45], FinnGen, a study from Finland, and the Copenhagen General Population Study and the Copenhagen City Heart Study (CGPS-CCHS)[46]. We had, in total, up to 40,042 cases (UKB: 12,377, GHS: 8032, SINAI: 409, MALMO: 2395, EstBB: 7629, HUNT: 3874; CPGS-CCHS: 1863; FinnGen: 3463) and 947,782 controls of European ancestry, and 5153 cases (UKB: 448, POAAGG: 3444, SINAI: 1261) and 21,650 controls of African ancestry in glaucoma analyses.

**Phenotype definition**. IOP in UKB was measured in each eye using the Ocular Response Analyzer (Reichert Corp., Buffalo, New York). Participants were excluded from this test if they reported having eye surgery in the preceding 4 weeks or having an eye infection. The Ocular Response Analyzer calculates two forms of IOP, a Goldmann-correlated IOP (IOPg) and a corneal-compensated IOP (IOPcc). IOPg most closely approximates the IOP measured by the Goldmann Applanation Tonometer (GAT), which has been the gold standard for measuring IOP, while IOPcc provides a measure of IOP that is adjusted to remove the influence of corneal biomechanics[47]. For this study, we focused on IOPg as this measurement is the most comparable to IOP measurements in other cohorts, and herein IOPg will be referred to as IOP. IOP in POAAGG was measured using a GAT. In GHS, IOP measurements were obtained from several instruments including GAT, Tono-pen and I-Care, which are correlated with IOPg readings from the Ocular Response Analyzer[48]. For GHS individuals who were not prescribed any IOP medications, we used the median of all IOP measurements available. For individuals who had an IOP medication prescribed, we used the median of IOP measurements available preceding the start date for IOP medications (if available). Individuals for whom we did not have non-medicated IOP values were excluded from the IOP genetic analyses. For association analyses of IOP, we excluded individuals with: (1) a glaucoma diagnosis; (2) IOP measures that were more than 5 standard deviations away from the mean; (3) more than a 10-mmHg difference between both eyes. We derived a mean IOP measure between both eyes for each individual. IOP of only one eye was used in instances where IOP measures for both eyes were not available.

Details on glaucoma definition in each cohort are given in the Supplementary Methods. In brief, glaucoma cases in GHS, SINAI, MALMO, HUNT, EstBB, FinnGen (v.R3) and CGPS-CCHS were defined by the presence of an ICD-10 H40 diagnosis code in either outpatient or inpatient electronic health records. In UKB, glaucoma cases were defined as individuals with either an ICD-10 H40 diagnosis or self-reported glaucoma (UKB field ID: 6148 or 20002). In the POAAGG cohort, glaucoma cases and controls were classified based on an ophthalmic examination by glaucoma specialists, and glaucoma suspects were also included in the cases[42].

**Statistics and reproducibility**. High coverage whole exome sequencing and genotyping was performed at the Regeneron Genetics Center[49,50] as described in Supplementary Methods. We estimated the association with IOP and glaucoma of genetic variants or their gene burden using REGENIE v1.0.43[51] (UKB, GHS, MALMO, SINAI), SAIGE[52] (HUNT, EstBB, FinnGen) or logistic regression (CGPS-CCHS). Analyses were adjusted for age, age², sex, an age-by-sex interaction term, experimental batch-related covariates, and genetic principal components, where appropriate. Cohort-specific statistical analysis details are provided in Supplementary Methods. Results across cohorts were pooled using inverse-variance weighted meta-analysis. Details on the PheWAS analysis conducted in UKB and GHS were provided in Supplementary Methods. Western blotting and ELISA analyses were repeated on three independent biological replicates and data are presented as mean ± SEM. Technical replicates ($n = 3$) were run for the ELISA analysis. $P$ values were calculated by one-way ANOVA with Tukey's multiple comparison test for multiple groups analysis (Supplementary Data 1). A total of 12 eyes were used to test the effect of increasing mAngptl7 levels in mouse eyes and Student's $t$ test was used to calculate the significance of the resulting change in IOP. The IOP was measured on 33 WT, 41 *Angptl7* KO and 15 *Angptl7* Het mice and conventional outflow facility was measured on 4 WT and 7 *Angptl7* KO mice. Unpaired Student's $t$-test was used to calculate the statistical significance of the results between the different genotypes. For in vivo siRNA knockdown of mAngptl7, we used 8, 6, 6 and 5 mouse eyes for siRNA#3, siRNA#5, PBS-treated and Naïve controls, respectively. Statistical significance was calculated using one-way ANOVA with Dunnett's post hoc analysis (Supplementary Data 1).

**In vitro characterization**. HEK293 cells, derived within Regeneron, were cultured in DMEM media 4.5 g/L D-Glucose, (+) L-Glutamine, (–) Sodium Phosphate, (–) Sodium Pyruvate supplemented with 10% FBS and 1% Penicillin-Streptomycin-Glutamine (Invitrogen), at 37 °C in a humidified atmosphere under 5% $CO_2$. The day before transfection, HEK293 cells were seeded in OptiMEM supplemented with 10% FBS. After 24 h, the cells were transfected with FuGENE 6, and 10 µg of pcDNA 3.1(+) encoding the following proteins: ANGPTL7 WT, Gln175His, Arg177* and Trp188*. After 24 h, the media was changed with 2% FBS OptiMEM. The following day, the cells were collected in RIPA buffer, supplemented with protease and phosphatase inhibitors (BRAND) or TRIzol reagent (Invitrogen) for protein and RNA analysis, respectively. The supernatants were transferred to an Eppendorf tube and immediately flash frozen for downstream protein analysis. Western blot analysis was performed using a rabbit polyclonal antibody against ANGPTL7 at 1:1000 dilution (10396-1-AP ProteinTech), using standard procedures. ANGPTL7 was quantified by ELISA according to manufacturer's instructions (LS-F50425 Life Sciences). The cell lysates were diluted 1:1000. The supernatants were diluted 1:10,000. The ELISA plate was read at 450 nm via SpectraMax M4 plate reader (Molecular Devices).

Total RNA was extracted using TRIzol reagent (Invitrogen) and RNeasy kit (Qiagen) according to manufacturer's instructions and treated with RNase-free DNase I (Promega). cDNA was synthesized using Superscript VILO cDNA synthesis kit (Invitrogen). Taqman analysis was performed using TaqMan Fast Advanced Master Mix (Applied Biosystems) in a QuantStudio 6 Flex (Applied Biosystems) and commercially available primers and probes for ANGPTL7 (Hs00221727—Applied Biosystems) and GAPDH (Hs02786624_g1—Applied Biosystems).

**In vivo characterization**. All animal protocols were approved by the Institutional Animal Care and Use Committee in accordance with the Regeneron's Institutional Animal Care and Use Committee (IACUC) and the Association for Research in Vision and Ophthalmology (ARVO) Statement for the Use of Animals in Ophthalmic and Vision Research. *Angptl7⁻/⁻* mice, on 63% C57BL/6NTac and 37% 129SvEvTac background, were generated by Regeneron Pharmaceuticals using the VelociMouse® technology[53]. Heterozygous mice (*Angptl7⁺/⁻*) were bred to generate age-matched wild-type, het and KO littermates that were used for experimentation at 3–4 months of age (mixed gender). Ocular anatomy in these mice was characterized using optical coherence tomography. Detailed methods on generation and characterization of KO mice are provided in Supplementary Methods. For in vivo siRNA experiments, we used C57BL/6 J male mice, 3-4 months old, from Jackson Labs.

**IOP measurements**. Mice were anesthetized and IOP was measured in both eyes using a TonoLab rebound tonometer (Colonial Medical Supply, Franconia, NH) before the start of Angptl7 injection and every day afterwards for six days[54–56]. When testing *Angptl7* siRNAs, IOPs were measured in each eye before then start of experiment and then every week until end of study. IOP measurements for both eyes were completed within 3–5 min. Six correct single measurements were done on each eye to generate one IOP reading. We took five IOP readings for each eye and used the average of those readings at each time-point.

**Aqueous humor outflow facility measurements**. Aqueous humor outflow facility (C) was measured by using our constant flow infusion technique in live mice[55–58]. Mice were anesthetized by using a 100/10 mg/kg ketamine/xylazine cocktail. A

quarter to half of this dose was administered for maintenance of anesthesia as necessary. One to two drops of proparacaine HCl (0.5%) (Bausch + Lomb) were applied topically to both eyes for corneal anesthesia. The anterior chambers of both eyes were cannulated by using a 30-gauge needle inserted through the cornea 1–2 mm anteriorly to the limbus and pushed across the anterior chamber to a point in the chamber angle opposite to the point of cannulation, taking care not to touch the iris, anterior lens capsule epithelium, or corneal endothelium. Each cannulating needle was connected to a previously calibrated (sphygmomanometer, Diagnostix 700; American Diagnostic Corporation, Hauppauge, NY, USA) flow-through BLPR-2 pressure transducer (World Precision Instruments [WPI], Sarasota, FL, USA) for continuous determination of pressure within the perfusion system. A drop of genteal (Alcon) was also administered to each eye to prevent corneal drying. The opposing ends of the pressure transducer were connected via further tubing to a 1 ml syringe loaded into a microdialysis infusion pump (SP200I Syringe Pump; WPI). The tubing, transducer, and syringe were all filled with sterile DPBS (Gibco). Signals from each pressure transducer were passed via a TBM4M Bridge Amplifier (WPI) and a Lab-Trax Analog-to-Digital Converter (WPI) to a computer for display on a virtual chart recorder (LabScribe2 software; WPI). Eyes were initially infused at a flow rate of 0.1 µl/min. When pressures stabilized within 10–30 min, pressure measurements were recorded over a 5-min period, and then flow rates were increased sequentially to 0.2, 0.3, 0.4, and 0.5 µl/min. Three stabilized pressures at 3-minute intervals at each flow rate were recorded. C in each eye of each animal was calculated as the reciprocal of the slope of a plot of mean stabilized pressure as ordinate against flow rate as abscissa.

**Injection of Angptl7 protein and siRNA into mouse eyes**. A 33-gauge needle with a glass microsyringe (5-uL volume; Hamilton Company) was used for injections of Angptl7 protein/siRNA into mice eyes. For intravitreal injections, the eye was proptosed, and the needle was inserted through the equatorial sclera and into the vitreous chamber at an angle of approximately 45 degrees, taking care to avoid touching the posterior part of the lens or the retina. Angptl7 protein (catalog# 4960-AN-025; R&D Systems, Minneapolis, MN) or siRNA (from Alnylam Pharmaceuticals, Supplementary Methods) or PBS (1uL) was injected into the vitreous over the course of 1 minute. The needle was then left in place for a further 45 s (to facilitate mixing), before being rapidly withdrawn. siRNA sequences for all six probes tested are provided in Table 2. Before and during intracameral injections of Angptl7 protein, mice were anesthetized with isoflurane (2.5%) containing oxygen (0.8 L/min). For topical anesthesia, both eyes received one to two drops of 0.5% proparacaine HCl (Akorn Inc.). Each eye was proptosed and the needle was inserted through the cornea just above the limbal region and into the anterior chamber at an angle parallel to the cornea, taking care to avoid touching the iris, anterior lens capsule epithelium, or corneal endothelium. Up to 1 µL of Angptl7 protein or PBS was injected into each eye over a 30-s period before the needle was withdrawn. Only one injection was administered at day 0.

**Reporting summary**. Further information on research design is available in the Nature Research Reporting Summary linked to this article.

## Data availability

Source data underlying main figures are provided in Supplementary Data 1. Uncropped and unedited images of the blots that appear in the main article are provided in Supplementary Data 2. All whole exome sequencing, genotyping chip, and imputed sequence described in this report are publicly available to registered researchers via the UK Biobank data access protocol. Additional information about registration for access to the data is available at http://www.ukbiobank.ac.uk/register-apply/. Further information about the whole exome sequence is available at https://www.ukbiobank.ac.uk/media/kqjmdwfg/access_064-uk-biobank-exome-release-faq_v10.pdf. Detailed information about the chip and imputed sequence is available at: https://www.ukbiobank.ac.uk/media/cffi4mx5/ukb-genotyping-and-imputation-data-release-faq-v3-2-1.pdf. DiscovEHR and the University of Pennsylvania exome sequencing and genotyping data can be made available to qualified, academic, non-commercial researchers upon request via a Data Transfer Agreement with Geisinger Health System and University of Pennsylvania, respectively. Genetic data for the HUNT, CGPS-CCHS and Estonia cohorts may be made available by contacting the respective institutions. FinnGen R3 data can be accessed at: https://www.finngen.fi/. Regeneron materials described in this manuscript may be available to qualified academic researchers upon request through our portal (https://regeneron.envisionpharma.com/vt_regeneron/). In certain circumstances in which we are unable to provide a particular proprietary reagent, an alternative molecule may be provided that behaves in a similar manner. Additional information about how we share our materials can be obtained by contacting Regeneron's preclinical collaborations email address (preclinical.collaborations@regeneron.com).

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

## Acknowledgements

We would like to thank everyone who made this work possible. In particular: the UK Biobank team, their funders, the dedicated professionals from the member institutions who contributed to and supported this work, and most especially the UK Biobank participants, without whom this research would not be possible. The exome sequencing was funded by the UK Biobank Exome Sequencing Consortium (i.e., Bristol Myers Squibb, Regeneron, Biogen, Takeda, AbbVie, Alnylam, AstraZeneca and Pfizer). This research has been conducted using the UK Biobank Resource under application number 26041; EstBB, we thank all participants and staff of the Estonian biobank for their contribution to this research and the analytical work of EstBB was carried out in part in the High Performance Computing Center of the University of Tartu. L.M and K.K were supported by grants from the Estonian Research Council PRG184 and by the European Union through the European Regional Development Fund (Project No. 2014-2020.4.01.15-0012); We want to acknowledge the participants and investigators of the FinnGen study. We also thank the MyCode Community Health Initiative participants for taking part in the DiscovEHR collaboration. This research received funding from Regeneron Pharmaceuticals. JOB was supported by the National Eye Institute (#R01EY023557-01), Vision Research Core Grant (P30 EY001583), and NIH Grant (LM010098). Funds also come from the F.M. Kirby Foundation, Research to Prevent Blindness, The UPenn Hospital Board of Women Visitors, and The Paul and Evanina Bell Mackall Foundation Trust. The Ophthalmology Department at the Perelman School of Medicine and the VA Hospital in Philadelphia, Pennsylvania, also provided support. The funders had no role in study design, data collection and analysis, decision to publish, or preparation of the manuscript. The Trøndelag Health Study (The HUNT Study) is a collaboration between HUNT Research Center (Faculty of Medicine and Health Sciences, NTNU, Norwegian University of Science and Technology), Trøndelag County Council, Central Norway Regional Health Authority, and the Norwegian Institute of Public Health. The genotyping in HUNT was financed by the National Institutes of Health; University of Michigan; the Research Council of Norway; the Liaison Committee for Education, Research and Innovation in Central Norway; and the Joint Research Committee between St Olav's hospital and the Faculty of Medicine and Health Sciences, NTNU. S.S received funding from the Independent Research Fund, Denmark (Sapere Aude Research leader, grant number 9060-00012B). V.R.M.C received funding from R21EY028273-01A1 and the Lisa Dean Moseley Foundation Grant.

## Author contributions

All authors contributed to critical review of the manuscript for important intellectual content, and final approval of submission of the manuscript for publication. Conceptualization: K.P., G.P., A.B., C.R., and G.C. Data curation: D.L., M.N.C. Genetic Analysis: K.P., L.G., M.A.F., A.H.A., S.A.D.G., A.L., T.M.T., C.S., D.S., K.K., B.B., S.S., E.S., E.J. Funding acquisition: G.D.Y., G.R.A., A.B., J.M.B., C.R., L.M., O.M., S.S., K.H. Mouse experiments: G.P., B.P, S.W., H.Y., W.F., S.Z., J.S.R., W.F. siRNA design and development: S.L., J.M., S.Waldron, S.H. In vitro experiments: G.D.G., T.P., L.M., M.D.S. Project administration: E.C., M.B.J. Datasets: J.M.B., V.R.M.C., H.V.G., R.S., B.B., K.H., B.O.A., C.W., K.K., L.M,. S.S., A.S., A.T,. O.M. Supervision: M.A.F., G.D.G., Y.H., K.H., L.M., K.K., A.E., J.W., A.B., C.R., G.C. Writing—original draft: K.P., G.P., G.D.G., T.V.Z., C.R., G.C. All authors contributed to securing funding, study design and oversight. All authors reviewed the final version of the manuscript (RGC Management and Leadership Team). C.B., C.F., A.L., and J.D.O. performed and are responsible for sample genotyping. C.B, C.F., E.D.F., M.L., M.S.P., L.W., S.E.W., A.L., and J.D.O. performed and are responsible for exome sequencing. T.D.S., Z.G., A.L., and J.D.O. conceived and are responsible for laboratory automation. M.S.P., K.M., R.U., and J.D.O. are responsible for sample tracking and the library information management system. X.B., A.H., O.K., A.M., S.O., R.P., T.P., A.R., W.S. and J.G.R. performed and are responsible for the compute logistics, analysis and infrastructure needed to produce exome and genotype data. G.E., M.O., M.N. and J.G.R. provided compute infrastructure development and operational support. S.B., S.Ba, S.C., K.S., S.K., and J.G.R. provided variant and gene annotations and their functional interpretation of variants. E.M., R.L., B.B., A.B., L.H., J.G.R. conceived and are responsible for creating, developing, and deploying analysis platforms and computational methods for analyzing genomic data. All authors contributed to the clinical informatics of the project (Clinical Informatics). All authors contributed to the analytical analysis of the project (Translational and Analytical Genetics). All authors contributed to the management and coordination of all research activities, planning and execution. All authors contributed to the review process for the final version of the manuscript (Research Program Management).

## Competing interests

The authors declare the following competing interests: Regeneron authors receive salary from and own options and/or stock of the company. C.W.'s spouse is an employee of the Regeneron Genetics Center. The remaining authors declare no competing interests.

## Additional information

[1]Regeneron Genetics Center, Tarrytown, NY 10591, USA. [2]Regeneron Pharmaceuticals, Inc., Tarrytown, NY 10591, USA. [3]K.G. Jebsen Center for Genetic Epidemiology, Department of Public Health and Nursing, NTNU, Norwegian University of Science and Technology, 7030 Trondheim, Norway. [4]HUNT Research Center, Department of Public Health and Nursing, NTNU, Norwegian University of Science and Technology, 7600 Levanger, Norway. [5]Clinic of Medicine, St. Olavs Hospital Trondheim University Hospital, 7030 Trondheim, Norway. [6]Estonian Genome Centre, Institute of Genomics, University of Tartu, Tartu, Estonia. [7]Department of Endocrinology, Clinic of Medicine, St. Olavs Hospital, Trondheim University Hospital, 7030 Trondheim, Norway. [8]Department of Ophthalmology, Pearlman School of Medicine, University of Pennsylvania, Philadelphia, PA, USA. [9]Alnylam Pharmaceuticals, Inc, Cambridge, MA 02142, USA. [10]Bayer AG, Pharmaceuticals, Research and Development, 13342 Berlin, Germany. [11]Department of Clinical Biochemistry, Rigshospitalet, Copenhagen University Hospital, Copenhagen, Denmark. [12]Division of Cardiovascular Medicine, Department of Internal Medicine, University of Michigan, Ann Arbor, MI, USA. [13]Department of Computational Medicine and Bioinformatics, University of Michigan, Ann Arbor, MI, USA. [14]Department of Human Genetics, University of Michigan, Ann Arbor, MI, USA. [15]Lund University, Department of Clinical Sciences Malmö, Malmö, Sweden. [16]Skåne University Hospital, Department of Emergency and Internal Medicine, Malmö, Sweden. *Lists of authors and their affiliations appear at the end of the paper. ✉email: aris.baras@regeneron.com; carl.romano@regeneron.com; giovanni.coppola@regeneron.com

## Regeneron Genetics Center

**RGC Management and Leadership Team** Goncalo R. Abecasis[1], Aris Baras[1✉], Michael Cantor[1], Giovanni Coppola[1✉], Andrew Deubler[1], Aris Economides[1], Luca A. Lotta[1], John D. Overton[1], Jeffrey G. Reid[1], Alan Shuldiner[1], Katia Karalis[1] & Katherine Siminovitch[1]

**Sequencing and Lab Operations** Christina Beechert[1], Caitlin Forsythe[1], Erin D. Fuller[1], Zhenhua Gu[1], Michael Lattari[1], Alexander Lopez[1], John D. Overton[1], Thomas D. Schleicher[1], Maria Sotiropoulos Padilla[1], Louis Widom[1], Sarah E. Wolf[1], Manasi Pradhan[1], Kia Manoochehri[1] & Ricardo H. Ulloa[1]

**Genome Informatics** Xiaodong Bai[1], Suganthi Balasubramanian[1], Suying Bao[1], Boris Boutkov[1], Siying Chen[1], Gisu Eom[1], Lukas Habegger[1], Alicia Hawes[1], Shareef Khalid[1], Olga Krasheninina[1], Rouel Lanche[1], Adam J. Mansfield[1], Evan K. Maxwell[1], Mona Nafde[1], Sean O'Keeffe[1], Max Orelus[1], Razvan Panea[1], Tommy Polanco[1], Ayesha Rasool[1], Jeffrey G. Reid[1], William Salerno[1] & Kathie Sun[1]

**Clinical Informatics** Amelia Averitt[1], Nilanjana Banerjee[1], Michael Cantor[1], Dadong Li[1], Sameer Malhotra[1], Deepika Sharma[1], Jeffery C. Staples[1] & Ashish Yadav[1]

**Translational and Analytical Genetics** Goncalo R. Abecasis[1], Joshua Backman[1], Amy Damask[1], Lee Dobbyn[1], Manuel Allen Revez Ferreira[1], Arkopravo Ghosh[1], Christopher Gillies[1], Lauren Gurski[1], Eric Jorgenson[1], Hyun Min Kang[1], Michael Kessler[1], Jack Kosmicki[1], Alexander Li[1], Nan Lin[1], Daren Liu[1], Adam Locke[1], Jonathan Marchini[1], Anthony Marcketta[1], Joelle Mbatchou[1], Arden Moscati[1], Charles Paulding[1], Carlo Sidore[1], Eli Stahl[1], Kyoko Watanabe[1], Bin Ye[1], Blair Zhang[1] & Andrey Ziyatdinov[1]

**Research Program Management** Esteban Chen[1], Marcus B. Jones[1], Michelle G. LeBlanc[1], Jason Mighty[1], Lyndon J. Mitnaul[1], Nirupama Nishtala[1] & Nadia Rana[1]

## GHS-RGC DiscovEHR Collaboration

Lance J. Adams[17], Jackie Blank[17], Dale Bodian[17], Derek Boris[17], Adam Buchanan[17], David J. Carey[17], Ryan D. Colonie[17], F. Daniel Davis[17], Dustin N. Hartzel[17], Melissa Kelly[17], H. Lester Kirchner[17], Joseph B. Leader[17], David H. Ledbetter[17], J. Neil Manus[17], Christa L. Martin[17], Raghu P. Metpally[17], Michelle Meyer[17], Tooraj Mirshahi[17], Matthew Oetjens[17], Thomas Nate Person[17], Christopher Still[17], Natasha Strande[17], Amy Sturm[17], Jen Wagner[17] & Marc Williams[17]

[17]Geisinger, Danville, PA 17821, USA.

## Estonian Biobank Research Team

Andres Metspalu[6], Mari Nelis[6], Reedik Mägi[6] & Tõnu Esko[6]

