## [Peer Review File · Communications Biology]

Reviewers' comments:

Reviewer #1 (Remarks to the Author):

The manuscript "Genetic and functional characterization of ANGPTL7 as a therapeutic target for glaucoma" by Kavita Praveen and co-authors aims to identify physiological modulators of IOP using genome- and exome-wide association analysis.

In this study samples from over >129,000 individuals were analyzed. The research consortium identified rare coding variants in ANGPTL7. Then, Angptl7 knockout mice were established, which have lower IOP. In addition, acute gene silencing via siRNA knockdown of Angptl7 in mice lowered IOP. The authors conclude that ANGPTL7 important for IOP homeostasis. They assume this could be a novel therapeutic target for glaucoma.

This is a well conducted study on a relevant topic. The manuscript is well written, but some improvements could be made.

ANGPTL7 was identified in a publication from 2020 using samples for UK Biobank (and FinnGen. What are the differences of the genetic analysis conducted in this study? What is the novel aspect of the genetic analysis?

The authors are also using samples from the FinnGen study. Were some of these samples also included in the 2020 study?

How many IOP measurements were carried out per eye and age in the animal studies? Were means per eye calculated?

Figure 2B: there were only 2 eyes in the control group? This is not sufficient for a statistical analysis. N needs to be increased.

Revise photos in figure 5, they are hard to see.

Minor comment:

Layout: Supplementary methods description should be consistent in full justification.

Reviewer #2 (Remarks to the Author):

In this manuscript, Praveen et al have characterized the role of ANGPTL7 in IOP regulation and glaucoma. Utilizing human genetic approaches, they have identified associations between rare coding variants of ANGPTL2 and intraocular pressure (IOP). Specifically, their data points to a protective effect of loss of function variants of ANGPTL2 against high IOP and glaucoma. Furthermore, they have performed in vitro characterization of the ANGPTL7 variants and found that Gln175His and other truncated variants cause a defect in the extracellular secretion of mutant ANGPTL7. Finally, using mouse models, they show that 1) intracameral injection of recombinant ANGPTL7 in mice elevates IOP; 2) Conversely, Angptl7 knockout mice exhibit reduced basal IOP, and partial knockdown of Angptl7 with siRNA in the trabecular meshwork (TM) resulted in the lowering of IOP. Overall, their data suggests a role for ANGPTL7 as a physiological regulator of IOP as well as a glaucoma risk gene. The human genetic analysis is well done and makes use of multiple cohorts to show the association between ANGPTL2 variants and IOP regulation/glaucoma and its protective effect. The concerns are: 1) Measurement of IOP using Tonolab is not the most reliable, especially when a change in IOP is rather small. Outflow measurements would have added value and strengthened their conclusions. 2) Would it be possible to show that the intracamerally injected recombinant ANGPTL7 can localize to the trabecular meshwork (TM) or other relevant ocular tissues? Is there a reliable antibody against ANGPT7 that would allow visualization of the protein using immunohistochemistry? 3) Histological analysis or transmission electron microscopy of the ocular angle in mice lacking ANGPTL7 or those with elevated levels of ANGPTL7 could provide insight into potential structural

changes.

4) Would have been useful to know the status of the molecular changes in the TM due to Angptl7 deficiency (RNA-seq or levels of other ECM proteins).

Minor concerns

1) Some parts of the Method section in the supplemental material could be moved to the main manuscript (example, intracameral injection of siRNA and recombinant protein).

2) Though RNAscope is employed for in situ hybridization, =it would be useful to have some form of negative control, especially for the human tissue. Also, consider moving the in-situ data performed on Angptl7 knockout mice and showing it as part of the main figure.

Reviewer #3 (Remarks to the Author):

Giovanni Coppola et al. performed genome- and exome-wide association analysis in >129,000 individuals with IOP measurements to identify physiological modulators of IOP. They report the identification and functional characterization of rare coding variants (including loss-of-function variants) in ANGPTL7 associated with reduction in IOP and glaucoma protection. They further validated the human genetics findings in mice by establishing that Angptl7 knockout mice have lower (~2 mmHg) basal IOP compared to wild-type, with a trend towards lower IOP also in heterozygotes. Conversely, increasing mAngptl7 levels via injection into mouse eyes increases the IOP. They also show that acute gene silencing via siRNA knockdown of Angptl7 in adult mice lowers the IOP (~2-4 mmHg), reproducing the observations in knockout mice. This paper is interesting. However, I also have some major concerns listed below:

1.The genetic data and conclusion in this study is similar to the published paper entitled 'Rare protein-altering variants in ANGPTL7 lower intraocular pressure and protect against glaucoma' (PMID:32369491). There is no need to publish it again.

2.The novelty of this study is the functional study of Angptl7 gene in mice which links the expression of Angptl7 and IOP. Since Angptl7 is also expressed in corneal stroma and sclera, did the Angptl7 KO mice have any other phenotype besides lowering the IOP?

3.Since this study only focused on IOP, not glaucoma, the title and the discussion should be revised accordingly to distinguish the correlation and difference between them.

We thank the Reviewers for their comments and suggestions for improvement. We have addressed individual points below, in red. Within the main text, the changes made have been highlighted yellow or have been tracked.

Referee expertise:

Referee #1: mouse models of glaucoma

Referee #2: glaucoma genetics, mouse models

Referee #3: ophthalmology, genetics, glaucoma

Reviewers' comments:

Reviewer #1 (Remarks to the Author):

The manuscript “Genetic and functional characterization of ANGPTL7 as a therapeutic target for glaucoma” by Kavita Praveen and co-authors aims to identify physiological modulators of IOP using genome- and exome-wide association analysis.

In this study samples from over >129,000 individuals were analyzed. The research consortium identified rare coding variants in ANGPTL7. Then, Angptl7 knockout mice were established, which have lower IOP. In addition, acute gene silencing via siRNA knockdown of Angptl7 in mice lowered IOP. The authors conclude that ANGPTL7 important for IOP homeostasis. They assume this could be a novel therapeutic target for glaucoma.

This is a well conducted study on a relevant topic. The manuscript is well written, but some improvements could be made.

We thank the Reviewer for their comments and suggestions for improvement.

ANGPTL7 was identified in a publication from 2020 using samples for UK Biobank (and FinnGen. What are the differences of the genetic analysis conducted in this study? What is the novel aspect of the genetic analysis?

This work makes several important additions to the genetic association study reported by Tanigawa et al as outlined below. We have also incorporated these points into the discussion (page 14, line 279) in response to the Reviewer’s question.

1. Our paper uses data from exome sequencing while analyses in Tanigawa et al were conducted on genotyped variants. An advantage of our approach is that we were able to uncover several additional rare and ultra-rare variants in ANGPTL7 (61 variants in addition to Gln175His and Arg177*) and incorporate them into a burden test. We identified and reported several ultra-rare variants in ANGPTL7 that were not reported in Tanigawa et al, which were independently associated with reduced IOP. In summary, exome sequencing strengthens the case for ANGPTL7 as a gene important for IOP regulation.

2. In addition to UKB and FinnGen (reported in Tanigawa et al), we have included several cohorts in our analyses of ANGPTL7 variants. For IOP, we have included data from Geisinger (GHS) in addition to UKB, and shown a consistent trend of lower IOP in variant carriers in both datasets. For glaucoma analyses of Gln175His and Arg177* variants, we included six additional cohorts (GHS, SINAI, MALMO, Estonia, HUNT, Copenhagen). The addition of new cohorts is important to further validate the original genetic findings, and also to better estimate the effect size across different populations. Amongst the cohorts we have added, we observed a replication of the association of Gln175His with glaucoma in HUNT. The addition of more cohorts is particularly important for the Arg177* loss of function variant as in the Tanigawa et al paper this variant was only tested in one cohort (UKB) because it was not observed in FinnGen. We show the effect of this variant on IOP across UKB and GHS, and in glaucoma across a total of 7 cohorts.
3. Tanigawa et al focused on European and Finnish European individuals in their analysis. We have also interrogated ANGPTL7 association with IOP and glaucoma in African ancestry individuals by incorporating the POAAGG cohort, as well as African ancestry individuals from SINAI and UKB. The African ancestry population has the highest prevalence of glaucoma but is understudied in genetic analyses of the disease. In the AFR population, we have identified an additional loss-of-function variant in ANGPTL7, Trp188*. To the best of our knowledge, ours is the first report of an ANGPTL7 variation survey, and the presence of this variant, in the AFR population.
4. We also report the results of a phenome-wide association analysis of the ANGPTL7 aggregate of variants across 14,050 binary and 10,032 quantitative traits in UKB and GHS, which included several custom-derived ocular traits. Our analysis has uncovered associations with corneal traits, which had not previously been shown for ANGPTL7, and offer important insights related to the evaluation of ANGPTL7 as a potential therapeutic target for IOP lowering/glaucoma.

The authors are also using samples from the FinnGen study. Were some of these samples also included in the 2020 study?

Tanigawa et al used release 4 (R4) of FinnGen, we used release 3 (R3). R3 is a subset of R4 so all our 3,463 cases and 93,036 controls should be part of their analysis. We have indicated the version of FinnGen dataset used in the Methods section (line 454) and in the supplementary information (line 111).

How many IOP measurements were carried out per eye and age in the animal studies? Were means per eye calculated?

Six correct single Tonolab tonometer measurements were collected on each eye, resulting in one IOP reading. We took 5 IOP readings for each eye and used the average of those readings at each time-point. We added a sentence detailing this aspect in the Methods section (page 22, line 491).

Figure 2B: there were only 2 eyes in the control group? This is not sufficient for a statistical analysis. N needs to be increased.

We thank the Reviewer for this comment. We have increased the sample size in the control group to N=4 (see revised Figure 6 on page 51).

Figure 6:

Revise photos in figure 5, they are hard to see.

We have replaced the photos in Figure 5E (page 50) with ones that are clearer.

Figure 5:

E

Minor comment:

Layout: Supplementary methods description should be consistent in full justification.

Thank you, we have fixed this.

Reviewer #2 (Remarks to the Author):

In this manuscript, Praveen et al have characterized the role of ANGPTL7 in IOP regulation and glaucoma. Utilizing human genetic approaches, they have identified associations between rare coding variants of ANGPTL2 and intraocular pressure (IOP). Specifically, their data points to a protective effect of loss of function variants of ANGPTL2 against high IOP and glaucoma. Furthermore, they have performed in vitro characterization of the ANGPTL7 variants and found that Gln175His and other

truncated variants cause a defect in the extracellular secretion of mutant ANGPTL7. Finally, using mouse models, they show that 1) intracameral injection of recombinant ANGPTL7 in mice elevates IOP; 2) Conversely, Angptl7 knockout mice exhibit reduced basal IOP, and partial knockdown of Angptl7 with siRNA in the trabecular meshwork (TM) resulted in the lowering of IOP. Overall, their data suggests a role for ANGPTL7 as a physiological regulator of IOP as well as a glaucoma risk gene.

The human genetic analysis is well done and makes use of multiple cohorts to show the association between ANGPTL2 variants and IOP regulation/glaucoma and its protective effect. The concerns are:

- 1) Measurement of IOP using Tonolab is not the most reliable, especially when a change in IOP is rather small. Outflow measurements would have added value and strengthened their conclusions.

We thank the Reviewer for this comment, we measured the outflow facility in WT and KO animals using the constant flow infusion technique (described in Millar et al, 2011, *IOVS*). We observed a 21% increase in conventional outflow facility in KO mice compared to WT mice. We have added the outflow results as panel B in Figure 8 in the main text (page 53). These results are consistent with IOP lowering observed in KO mice.

Figure 8:

- 2) Would it be possible to show that the intracamerally injected recombinant ANGPTL7 can localize to the trabecular meshwork (TM) or other relevant ocular tissues? Is there a reliable antibody against ANGPTL7 that would allow visualization of the protein using immunohistochemistry?

This is a great point and we agree with the Reviewer this experiment would provide helpful information. Unfortunately, the commercial antibodies we have tested so far for IHC are not working on eye sections.

- 3) Histological analysis or transmission electron microscopy of the ocular angle in mice lacking ANGPTL7 or those with elevated levels of ANGPTL7 could provide insight into potential structural changes.

Another great point. We have performed H&E staining on eyes from ANGPTL7 KO and WT mice and we observed no differences in the ocular angle. We have added these results as a Supplementary Figure (Figure S7, page 27)

Figure S7:

- 4) Would have been useful to know the status of the molecular changes in the TM due to Angptl7 deficiency (RNA-seq or levels of other ECM proteins).

This is an active area of investigation in our group, as we agree it would help to better understand the biology and role of Angptl7 in IOP regulation. In the current study we were not able to perform RNAseq studies as Angptl7 KO mice were derived from a mixed strain background, which would have made the interpretation of gene expression data difficult. We are rederiving the animals on a pure B6 background, and plan to perform this work on this rederived strain.

Minor concerns

- 1) Some parts of the Method section in the supplemental material could be moved to the main manuscript (example, intracameral injection of siRNA and recombinant protein).

We thank the Reviewer for this suggestion, we have moved the description of injection of siRNA and recombinant protein into the main text methods and also included the outflow measurement methods in the main text.

- 2) Though RNAscope is employed for in situ hybridization, =it would be useful to have some form of negative control, especially for the human tissue. Also, consider moving the in-situ data performed on Angptl7 knockout mice and showing it as part of the main figure.

We have added a negative and a positive control for the in situ as a Supplemental Figure (Figure S6, page 26). We also moved the in situ data on Angptl7 KO and WT mice into the main text as Figure 7 (page 52).

Figure S6:

Figure 7:

Reviewer #3 (Remarks to the Author):

Giovanni Coppola et al. performed genome- and exome-wide association analysis in >129,000 individuals with IOP measurements to identify physiological modulators of IOP. They report the identification and functional characterization of rare coding variants (including loss-of-function variants) in ANGPTL7 associated with reduction in IOP and glaucoma protection. They further validated the human genetics findings in mice by establishing that Angptl7 knockout mice have lower (~2 mmHg) basal IOP compared to wild-type, with a trend towards lower IOP also in heterozygotes. Conversely, increasing mAngptl7 levels via injection into mouse eyes increases the IOP. They also show that acute gene silencing via siRNA knockdown of Angptl7 in adult mice lowers the IOP (~2-4 mmHg), reproducing the observations in knockout mice. This paper is interesting. However, I also have some major concerns listed below:

1.The genetic data and conclusion in this study is similar to the published paper entitled 'Rare protein-altering variants in ANGPTL7 lower intraocular pressure and protect against glaucoma' (PMID:32369491). There is no need to publish it again.

The genetic data in our paper extends the analysis by Tanigawa et al in the ways outlined in response to Reviewer #1. Also, as the Reviewer notes in the next point, we extend our genetic results to functional studies and animal models. We believe it would be beneficial to share these results with the scientific community.

2.The novelty of this study is the functional study of Angptl7 gene in mice which links the expression of Angptl7 and IOP. Since Angptl7 is also expressed in corneal stroma and sclera, did the Angptl7 KO mice have any other phenotype besides lowering the IOP?

We did not observe any ocular changes on anterior segment coherence tomography and, importantly, we also did not observe a difference in corneal thickness between wild-type and Angptl7 KO mice (text to this effect is given on page 11, line 237 in the main text and Figures S7 and S8). No other ocular phenotypes were observed in Angptl7 KO mice.

3.Since this study only focused on IOP, not glaucoma, the title and the discussion should be revised accordingly to distinguish the correlation and difference between them.

While the functional studies assess changes in IOP, we submit that a focus on the effect of ANGPTL7 on glaucoma is warranted for the following reasons: 1. The human genetic analyses indicate that the variants in ANGPTL7 that associate with lower IOP also associate with protection from glaucoma. This protection from glaucoma was crucial to establish because – due to the susceptibility of IOP measurements to corneal dynamics – not all variants associated with IOP changes will be associated with glaucoma risk or protection. 2. The beneficial effect of lowering IOP on slowing glaucoma progression or delaying onset has been established (Hejil et al, 2002, *Clinical Sciences*). IOP is the endpoint used in clinical trials of glaucoma medications. Based on current understanding and standards in the field, and in the light of our human genetics findings, we feel that extrapolating from mouse studies showing that ANGPTL7 inhibition can lower IOP to protection from glaucoma is reasonable. However, we have also adjusted the title of the paper to specify ANGPTL7 as a potential IOP-lowering target for glaucoma. We also discuss in depth in the manuscript the effect of ANGPTL7 as a regulator of IOP homeostasis thus emphasizing that the human genetic data showing protection from glaucoma is likely driven by IOP modulation.

REVIEWERS' COMMENTS:

Reviewer #1 (Remarks to the Author):

The authors carefully revised the manuscript and addressed all previous comments. I do not have any further comments. Nice work!

Reviewer #2 (Remarks to the Author):

The authors have addressed all the major questions that were raised by the reviewers. The inclusion of new data has strengthened the manuscript.

Overall, their data support their conclusions.

Reviewer #3 (Remarks to the Author):

Now I am satisfied with the author's revision